# Sulfur and chlorine budgets control the ore fertility of arc magmas

Carter Grondahl [1] & Zoltán Zajacz [1,2 ✉]

Continental arc magmas supply the ore-forming element budget of most globally important porphyry-type ore deposits. However, the processes enabling certain arc segments to preferentially generate giant porphyry deposits remain highly debated. Here we evaluate the large-scale covariation of key ore-forming constituents in this setting by studying silicate melt inclusions in volcanic rocks from a fertile-to-barren segment of the Andean Southern Volcanic Zone (33–40 °S). We show that the north-to-south, fertile-to-barren gradient is characterized by a northward increase in S and Cl concentrations and a simultaneous decrease in Cu. Consequently, we suggest that the concentration of S and Cl rather than the concentration of ore metals regulates magmatic-hydrothermal ore fertility, and that the loss of volatiles prior to arrival in the upper crust impacts ore-forming potential more than magmatic sulfide saturation-related ore metal scavenging.

---

[1] Department of Earth Sciences, University of Toronto, Toronto, Canada. [2] Present address: Department of Earth Sciences, University of Geneva, Geneva, Switzerland. ✉email: zoltan.zajacz@unige.ch

Porphyry-type ore deposits are major contributors to the global supply of Cu, Mo, Au, and Ag. Large deposits typically form beneath subduction-related volcanoes on thick continental crust. The key constituents for their genesis, ore metals and S, are both supplied by magma-derived aqueous fluids[1]. Furthermore, the transport of many ore metals is tied to the availability of Cl as a ligand, with which they form aqueous complexes[2–4]. Nevertheless, the budget of ore metals, S and Cl in arc magmas is poorly constrained. In particular, the roles of magma differentiation in lower crustal hot zones and magmatic sulfide saturation are highly debated[5–12].

Much of our current understanding of continental arc magma fertility is based on whole-rock Cu concentrations. The inverse relationship between Cu concentration and crustal thickness and the degree of magma differentiation led several researchers to suggest that chalcophile elements such as Cu and Au are sequestered in lower crustal sulfide accumulations during magma ascent and differentiation[5–7]. The creation of unusually metal-rich magma batches by remobilizing the metal budget of previously fractionated sulfides has been proposed as a solution to the apparent paradox that the tectonic setting most likely to form porphyry Cu deposits preferentially produces magmas with relatively low Cu concentrations[5,7,9]. In contrast to this paradigm of punctuated magmatic fertility, others note that even relatively Cu-poor magmas (i.e. <75 ppm Cu) contain sufficient Cu to allow the generation of large porphyry deposits[13], and instead emphasize a gradual evolution to volatile-rich magmas occurring at deep crustal levels[8,11,12]. Importantly, an arc-scale assessment of magmatic volatile budgets on ore fertility is currently missing.

To address this, we present the first large-scale systematic study that simultaneously investigates the budget of S, Cl, and a broad range of ore metals in continental arc magmas by using silicate melt inclusions (SMI) in minerals. We sampled mafic-intermediate eruptive products from seven modern volcanoes (1–4 samples each) between 33–40 °S in the Andean Southern Volcanic Zone (SVZ; Fig. 1 and Supplementary Fig. 1), which is an important natural laboratory often used to explore the origin of subduction-related geochemical fingerprints[14–18]. Major trends through this segment include south-to-north crustal thickening from ~35 to ~50 km in conjunction with an increasingly compressional stress regime. This is attended by a systematic shift towards more diverse and on average more silicic magma compositions[14,19,20]. Importantly, the south-to-north tectono-magmatic trends exhibited by the modern SVZ closely resemble those seen as a function of time in the Miocene-Pliocene Teniente Volcanic Complex which culminated in the formation of El Teniente, one of the world's two largest porphyry Cu-Mo deposits (see Supplementary Discussion)[15].

The studied SMI are microscopic-sized droplets of quenched silicate melt enclosed in igneous olivine and pyroxene phenocrysts, which preserve a unique record of the abundance of volatile ore-forming constituents free from the effects of syn-eruptive magma degassing. An additional advantage is that SMI may collectively preserve a record of the composition of various deeper-sourced magma batches which amalgamate in upper crustal magma reservoirs prior to eruption[22–26]. Consequently, the investigation of SMI in the eruptive products of a specific volcano can yield insights into processes operating prior to the arrival of magmas in the upper crust. Such information is unavailable via traditional whole-rock analyses, because the compositional differences between various magma batches are typically eradicated by magma mixing and mingling processes.

We used laser ablation inductively coupled plasma mass spectrometry (LA-ICP-MS) and electron microprobe analysis

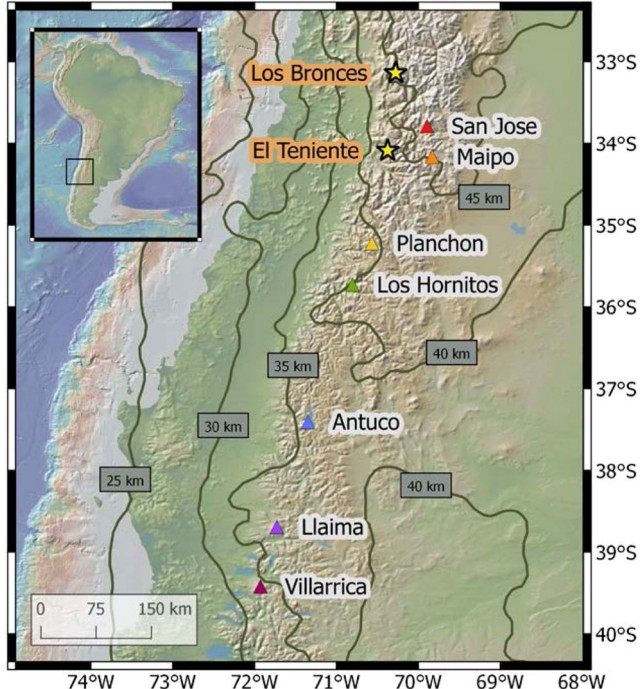

**Fig. 1 Locations of the sampled volcanoes (coloured triangles).** All are major volcanic centres except for Los Hornitos, which are small monogenetic basaltic cones just south of Cerro Azul, and erupted among the most primitive material in the studied arc segment. Yellow stars show the location of two recent (ca. 5 Ma) world-class porphyry Cu-Mo deposits formed just prior to migration of the northern SVZ to its current location. Labelled contours show the northward increase in depth to the MOHO, using data from ref. [19]. Sample compositions in the context of previous studies on these volcanoes are shown in Supplementary Fig. 1. Primitive magmatism in the SVZ is described in detail by ref. [21].

(EPMA) to simultaneously determine the concentrations of S, Cl, and ore metals, including the ultra-trace level (<0.1 ppm) but highly informative Ag, Au, and Pt. We focus on the ore-forming constituent endowment of arc magmas upon arrival in upper crustal magma reservoirs, and therefore present only SMI with $SiO_2 \le$ ~60 wt%, nearly all of which are olivine-hosted ($n = 133$; Mg# 72–88; Mg# = molar $[Mg/(Mg + Fe)]$ * 100) except for a few clinopyroxene- ($n = 6$; Mg# 78–85) and orthopyroxene-hosted ($n = 2$; Mg# 73–76) SMI. Within this dataset, we first look at the most primitive SMI from each location ('pSMI', $n = 67$), filtered for low $K_2O$ and high host Mg# to obtain the closest available approximation of initial mantle-derived magma compositions (Supplementary Table 1). Second, we examine trends in the full SMI dataset ('fSMI', $n = 141$) to explore how the prevailing conditions of early magma differentiation at lower to mid-crustal depths affect ore fertility.

We show that magmas in the fertile north arrive in the upper crustal ore-forming environment with a moderately depleted budget of chalcophile ore metals due to magmatic sulfide fractionation in deeper crustal magma reservoirs, whereas the barren southern magmas preserve their ore metal budget. Magmas in the north are, however, S and Cl rich, unlike those to the south, which lose a large fraction of their initially moderate S and Cl budget at deeper crustal levels during early differentiation and ascent. We conclude that S and Cl availability is the key factor regulating magmatic fertility, and that strong enrichments in magmatic ore metal concentrations are not a prerequisite for porphyry ore genesis.

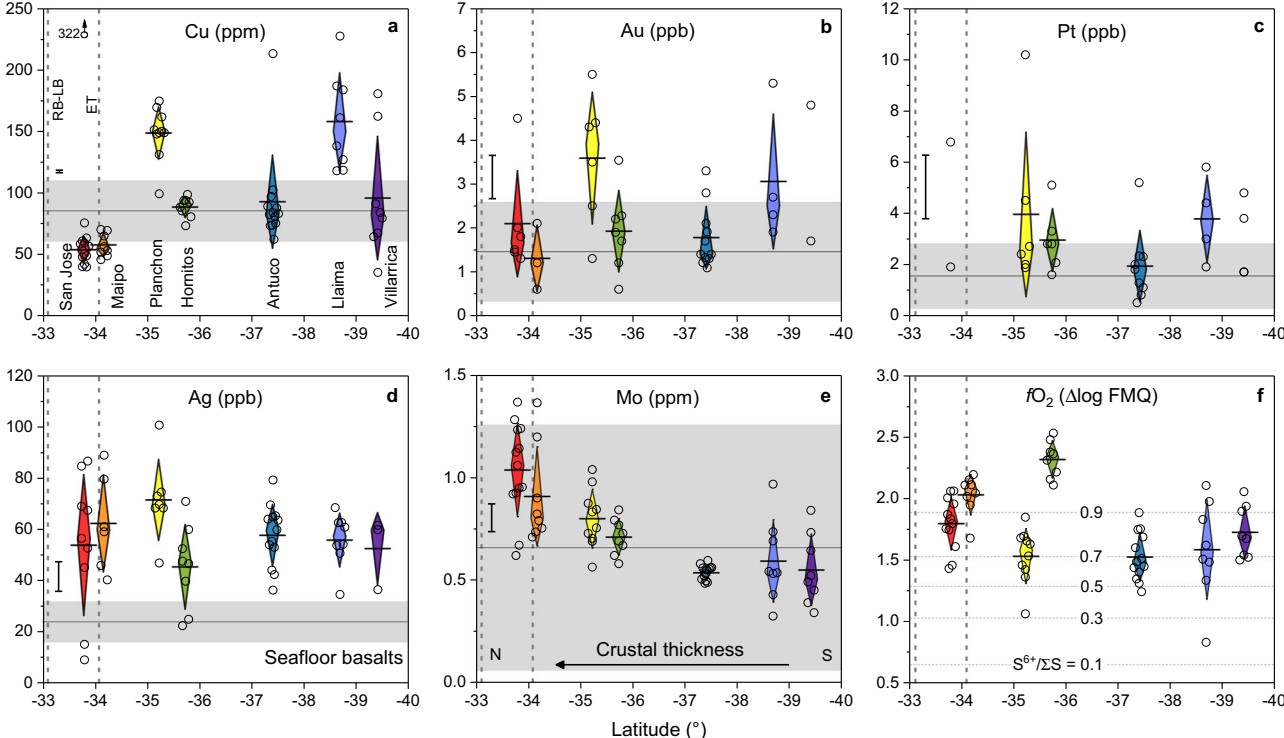

**Fig. 2 The variation of pSMI ore metal concentrations and the redox state of host magmas as a function of latitude. a–e** The concentration of the ore metals indicated at the top of each diagram. **f** Oxygen fugacity in log units relative to the Fayalite-Magnetite-Quartz (FMQ) oxygen buffer based on V partitioning between olivine and SMI[28]. On all panels, diamond lengths correspond to 1σ around the arithmetic mean (horizontal line), whereas diamond widths are centered on the median value. Vertical bars on the left margin display a representative 1σ uncertainty on the individual data points (open circles). Vertical reference lines RB-LB and ET show the latitudes of the world-class Rio Blanco-Los Bronces and El Teniente porphyry Cu deposits, respectively. Horizontal lines and grey fields on **a–e** indicate the average and 1σ range of global seafloor basalts, respectively[27], and the contour lines on **f** indicate the $S^{6+}$/total S ratio in the silicate melt as a function of $fO_2$ at an appropriate temperature (1100 °C; ref. [29]). Sulfide saturation-related scavenging of Cu, Au, and Pt occurs during crustal thickening, despite an accompanying shift to slightly more oxidizing conditions. The outlying behaviour of Planchon (**a**, **b**) compared to San Jose and Maipo suggests that any sulfide scavenging of ore metals occurred to a lesser extent here. This is plausibly related to relatively high Fe concentrations at Planchon and therefore increased sulfide solubility in the silicate melt[30].

## Results and discussion

**Ore-forming element concentrations**. The pSMI compositions show that the concentrations of ore metals vary through the SVZ (Fig. 2a–e). Average Cu concentrations, as well as the range in concentration, increase from the north to the south (40–75 ppm at San Jose and Maipo vs 80–200 ppm south of Maipo). Based on their chalcophile behaviour, Au and Pt would be expected to show patterns of depletion in a manner similar to Cu. Full understanding of variations in Au and Pt budgets is obscured by higher analytical uncertainties due to the ultra-trace level concentrations of these elements (1–6 ppb), although the latitudinal variation of the Au and Pt data is not inconsistent with that of Cu within error. Conversely, Mo is enriched in the north and Ag shows no variation with latitude. Compared to the much-studied seafloor basalt suites[27], the pSMI contain more Ag but comparable Mo, Au and Pt. Most seafloor basalt Cu concentrations are near 80 ppm, similar to the southern pSMI. In addition, northern magmas appear to be slightly more oxidized than those in the south and exhibit ~0.5 log units higher $fO_2$ (Fig. 2f). As opposed to the trends seen for most ore metals, volatile element concentrations in pSMI are notably higher in the north (2000–3000 ppm S and 1200–1500 ppm Cl) than the south (<1500 ppm S and <1000 ppm Cl; Fig. 3).

**Ore-forming element behaviour during magma differentiation**. We use $K_2O$ concentration in the silicate melt as a proxy of

magma differentiation because it is a sensitive tracer of both crystal fractionation and crustal contamination[14], and it is resistant to post-entrapment modification in silicate melt inclusions[35]. In addition, $K_2O$ and other highly incompatible elements in the SMI increase sharply over a narrow range of $SiO_2$ concentration and host mineral Mg# (see Supplementary Figs. 3 and 4), showing that the observed variation in melt composition develops at mid- to lower crustal depths on longer timescales facilitating significant assimilation of incompatible element rich partial melts of crustal materials while the concentrations of compatible major elements are at least partially buffered by the resident crystal mush[36,37]. Consistent with this proposition, the baseline geochemical characteristics of mafic SVZ magmas are widely accepted to originate prior to arrival in the upper crust (see Supplementary Discussion). This is supported by the presence of the characteristic latitudinal compositional trends in even the most primitive (e.g., deeply sourced) magma at each volcano (e.g., refs. [21,38–40]), and the difficulty of maintaining stable thermal conditions in upper crustal magma reservoirs permissive of prolonged storage and differentiation on the timescales and volumes inferred for arcs[36].

The variation of ore metal, S, and Cl concentrations as a function of $K_2O$ among fSMI demonstrate further contrast between the north and south. At Villarrica in the south, Cu increases, whereas S and Cl both decrease with increasing $K_2O$. In the north, however, Cu and S both decrease slightly with increasing $K_2O$, whereas Cl shows the opposite trend

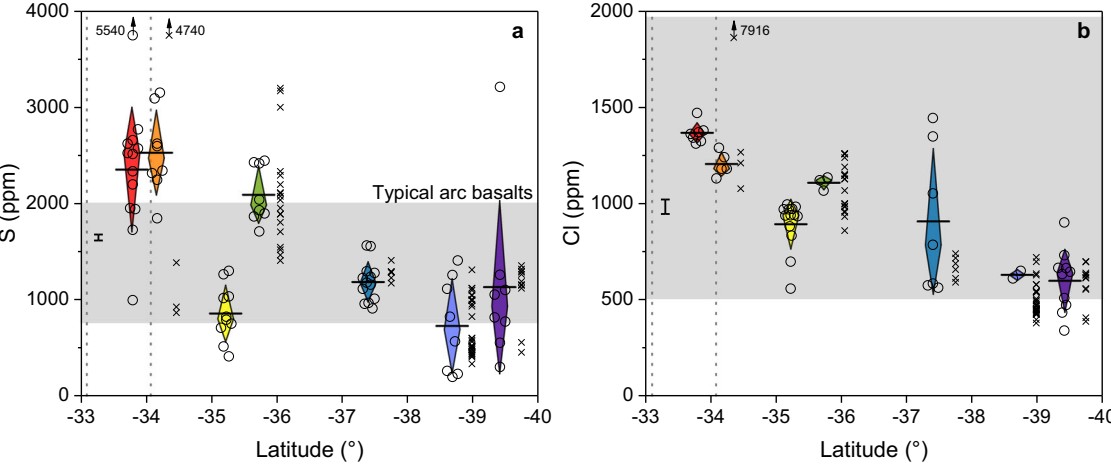

**Fig. 3 The variation of volatile element concentrations in pSMI as a function of latitude. a** The concentration of S. **b** The concentration of Cl. Symbols as in Fig. 2. Grey fields show typical arc basalt ranges[31].The relatively primitive S and Cl concentrations are lower in the south even before further loss during deep differentiation (Fig. 4). Chlorine was determined by EPMA (pSMI $n = 47$). The LA-ICP-MS and EPMA data are consistent with each other for S and other elements as well (Supplementary Fig. 2) and similar to previously reported data from the SVZ (small x symbols; pSMI filter applied)[32] with the key exception that northern S concentrations determined by LA-ICP-MS are higher than the corresponding EPMA data. This is due to post-entrapment paritioning of S into the bubble within the SMI[33,34], and highlights the advantage of LA-ICP-MS measurements which determine the bulk SMI composition (see 'Methods').

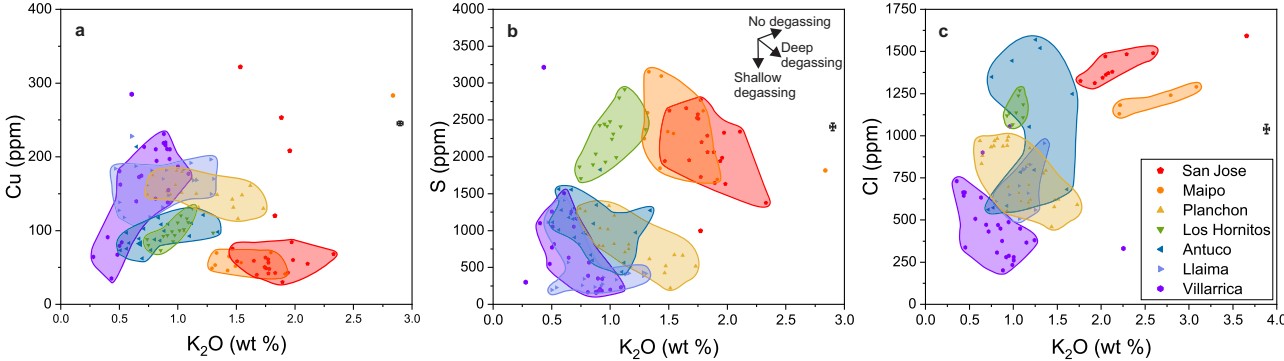

**Fig. 4 Variation of Cu, S and Cl with progressive magma differentiation. a** The variation of Cu concentrations as a function of $K_2O$, used as an indicator of magma differentiation. **b** The variation of S concentration. **c** The variation of Cl concentration. Data on all panels are from fSMI, which includes pSMI. Crosses along the right border show representative $1\sigma$ uncertainties for an intermediate concentration range (often smaller than the data symbols). Data in **c** were determined by EPMA (fSMI $n = 102$). Although sulfide fractionation is limited or absent in southern magmas (positive $Cu$-$K_2O$ trends and/or high Cu concentrations), they lose S and Cl during deep differentiation, effectively limiting the ore metal-transport capacity of magmatic fluids released in the upper crust. Northern magmas retain sufficient S and Cl to facilitate the fluid-mediated transport of the available ore metals to the site of ore formation and their subsequent precipitation. The lack of clear trends at Antuco and Llaima on **c** are likely due to more variability in magma storage depth at these volcanoes. Planchon S and Cl (e.g., degassing) behaviour is similar to Villarrica, although it appears that some minor sulfide saturation counterbalanced Cu build-up during differentiation. Note that Planchon moved far less than Maipo- and San Jose-latitude magmatism during previous arc migrations[15]. It therefore likely overlies a relatively well-established magmatic plumbing system reminiscent of the more southern volcanoes; Planchon is the northern-most SVZ stratovolcano to recently erupt basalt. Unlike the other volcanoes, the positive trends at the primitive monogenetic Los Hornitos cones (**a–c**) are consistent with the more oxidized (i.e., no magmatic sulfide fractionation; Fig. 2a, f) and undegassed magma compositions observed here.

(Fig. 4). We explain these observations by combining models of mantle melting and contrasting lower crustal differentiation processes.

Southern magmas ascend through relatively thin crust and therefore the reduced opportunity for crustal processing should allow their pSMI to provide closer approximations of primitive mantle melts than those from northern magmas. The least differentiated southern pSMI ore metal concentrations can be reproduced by relatively simple mantle melting models (see 'Methods') at melting extents previously proposed for magma genesis in the SVZ[40], and without invoking any transfer of ore metals from the subducted slab to the mantle wedge (Fig. 5).

The general lack of Cu depletion with progressing magma differentiation south of Maipo precludes extensive magmatic sulfide saturation (Fig. 4a). Therefore, the observed (and implied) decrease in S and Cl concentrations as differentiation proceeds within individual volcanoes (Fig. 4) can be explained by the following two processes that likely act simultaneously, although with variable influence.

The first is increased degassing at lower to mid crustal depths, which may allow efficient Cl partitioning into the fluid even from relatively mafic melts[41]. The fluid/silicate melt partition coefficient of Cl has been shown to strongly increase with increasing pressure between 0.05 and 0.5 GPa[41–43] and the extrapolation of this trend

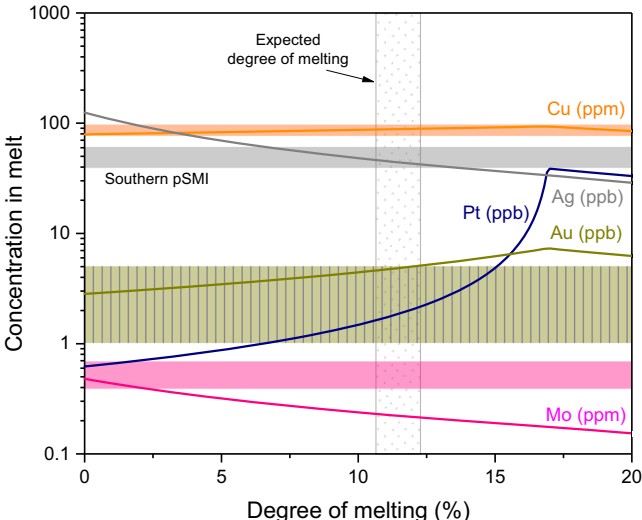

**Fig. 5 Ore metal concentrations from a mantle non-modal batch melting model (lines) compared to the measured concentrations in southern pSMI (coloured fields).** The measured concentrations are consistent with the modelled concentrations at the previously constrained degree of partial melting (patterened field)[40], though Mo is likely added to the magma prior to ascent (Supplementary Fig. 3). See 'Methods' for a detailed explanation of this model.

allowed us to estimate that the degassing of about 2 wt% aqueous fluid at $P = 1$ GPa would be required to explain the observed drop in Cl concentrations at Villarrica (Fig. 4c, see 'Methods'). In contrast, degassing at upper crustal pressures could not lead to Cl depletion in the residual mafic-intermediate melt, because of the low fluid/melt partition coefficient of Cl at these conditions and compositions[44]. This extent of degassing also reproduces the observed decrease in S concentration, assuming S was present in both 2− and 6+ oxidation states which is consistent with the estimated $fO_2$ for these magmas (Fig. 2f)[44]. The loss of S by degassing also explains why sulfide saturation was avoided in the south. This degassing scenario is consistent with the relatively high volcanic volatile emissions in the south[45] and the fact that Villarrica lies above the projection of the Valdivia fracture zone, a slab feature with the capacity for enhanced serpentinization and therefore $H_2O$ transport to the mantle source of magmas[46]. Although $H_2O$ solubility is ≈20 wt% for basalts in the lower crust (at $P \approx 1$ GPa)[47], some degassing could still occur at greater than upper crustal depths because the comparatively low solubility of $CO_2$ will induce early degassing of mixed $H_2O$-$CO_2$ fluids as magma ascends[48]. Similar deep degassing of basaltic arc magmas has previously been invoked to explain the much lower than expected $CO_2$ concentrations in mafic SMI compared to estimates based on volcanic gas compositions[31,49]. For example, MELTS[50] modelling of a basaltic Villarrica pSMI (host Mg# 85) containing a relatively low plausible pre-degassing $CO_2$ concentration (4500 ppm, following differentiation from a typical sub-arc mantle melt)[31,49] results in volatile saturation already in the mid crust (~700 MPa). The molar $CO_2$ and $H_2O$ fractions in such a fluid would be ~40–60%, assuming 5–7 wt% $H_2O$ in the melt. In addition, the $H_2O$ concentrations in even primitive arc magmas may reach ≥6 wt% $H_2O$[31,49,51]—enough to saturate $H_2O$ by the mid-crust when accounting for the elevation in $H_2O$ concentrations accompanying the amount of crystallization needed to derive mafic basaltic andesite compositions. There may be a delicate balance for efficient Cl degassing, as recent experiments using rhyolitic melts have shown that $CO_2$ can hinder Cl partitioning into magmatic fluids at upper crustal pressures[52], although the

higher density of fluid degassed at greater depth may partially mitigate this. We currently lack the experimental data to properly model these specific scenarios.

The second responsible process for the decrease in S and Cl concentrations is the assimilation of plutonic roots[53–55]. As older magma batches likely lost most of their volatile budget upon complete solidification, the incorporation of their partial melts would dilute S and Cl while increasing both fluid-mobile (e.g., $K_2O$) and -immobile (e.g., Nb) incompatible elements (plotted against host Mg# and one another in Supplementary Fig. 4). The implied absence of significant magmatic sulfide fractionation during the ascent and differentiation of preceding magma batches means that no residual phase is withholding Cu during the partial melting and assimilation of these rocks and therefore the dilution effect does not apply to Cu.

The variable influence of these processes is apparent in Fig. 4, where data from different volcanoes display more variation and the simple fSMI $K_2O$-Cl trends are not as well developed at Antuco and Llaima as at Villarrica and Planchon, despite similar $K_2O$-S trends. We propose that Villarrica represents one end-member scenario, where even though magma generation and ascent rates are relatively high[10,46] and the degree of crustal assimilation is relatively low[56], degassing prior to arrival in the upper crust may still remove large proportions of the initial S and Cl budget. Based on the similarity of their S-Cl-$K_2O$ behaviour, Planchon may represent an expression of the Villarrica-type endmember within thicker crust. Evidence of these processes are obscured at Llaima and especially Antuco as local circumstances including variable post-glacial magma source and storage dynamics[57–60] likely introduce additional variation in the data. Notably, the positive Llaima $K_2O$-Cl trend departs from an intermediate position along the negative Villarrica trend. Assuming a similar initial $K_2O$/Cl to Villarrica based on the proximity of the two volcanoes, this suggests that Llaima too experienced deep Cl loss prior to additional pre-eruptive storage at higher levels in the crust.

Prior to any degassing and dilution by crustal assimilation, the primary volatile and fluid-mobile incompatible element (e.g., $K_2O$) endowment will be a function of the slab-derived element flux and the extent of mantle melting[32]. Therefore, the inter-volcano variability between the S, Cl and $K_2O$ concentrations preserved within pSMI will relate to local variability in the composition of the mantle source and the added slab derived component as well as the extent of melting. Furthermore, in the case of S, redox conditions at the time of mantle melting may also play an important role. However, these are unlikely explanations for the observed variation with progressing magma differentiation within individual volcanic systems. In particular, the strong decoupling between K and Cl within a volcanic system is unlikely to be a consequence of variability in the slab-derived element flux. Therefore, the observed intra-volcano trends were likely established after mantle melting but prior to arrival in the upper crust, leaving assimilative dilution and degassing during lower to mid crustal storage and/or ascent as plausible causes. While SMI populations represent mixing of variably degassed/diluted magma batches, the Cl depletion signatures in these magma batches must have been established prior to their emplacement and amalgamation in upper crustal magma reservoirs.

**Lower-crustal magmatic sulfide fractionation.** In the fertile SVZ north of Planchon, deep crustal magma differentiation affects the budget of ore forming elements in a contrasting manner. Here, the lower Cu and Au concentrations in the pSMI reflect sulfide saturation in the lower crust. The lack of Ag depletion and the at most moderate drop of Au concentrations relative to the southern

magmas imply the fractionation of dominantly monosulfide solid solution (MSS) rather than a liquid sulfide phase[61,62].

It is challenging to determine the precise amount of S lost during magma differentiation in the lower crust by either sulfide fractionation or degassing. This is a consequence of transitions between the multiple possible valence states of S, which influence the S concentration in the silicate melt at sulfide saturation[63] and the fluid/silicate melt partition coefficients of S[44]. However, the amount of sulfide fractionated in the north can be estimated by model calculations relying on sulfide solubility models applied to primitive mantle-derived basalts and the silicate melts in the lower crust represented by the pSMI, the degree of fractional crystallization required to yield the pSMI compositions from a primitive mantle melt, and the difference between measured and modelled pSMI Cu concentrations. Deep degassing is not as prominent here as in the south, because Cl concentrations increase with progressing differentiation (Fig. 4c) despite the higher fluid/melt partition coefficient of Cl at elevated pressure[41]. This is consistent with the proposition that magma differentiation takes place under relatively high pressure (1–1.5 GPa) in the north resulting in elevated $H_2O$ and $CO_2$ solubility in the silicate melt[48]. We estimate that ascending magmas fractionate $\sim 1300 \pm 720$ ppm $S^{2-}$ as iron sulfide, equivalent to $\sim 0.36 \pm 0.20$ wt% FeS, with a solid:liquid sulfide ratio of $\sim 0.97{:}0.03$ constrained by the observed Cu/Au ratio (and within error of the Cu/Ag ratio) in the silicate melt. This bulk sulfide would scavenge $\sim 60$–70% of the Cu and Au and $\sim 10$% of the Ag from the initial budget of the primitive magma (Supplementary Table 3). This model is consistent with both the observed sulfur concentrations in the SMI and the estimated $fO_2$ values (see 'Methods'), and provides corroborating evidence for the importance of deep, MASH-type processes in affecting the ore metal endowment of magmas in fertile arc segments (see also Supplementary Fig. 3). Yet, as apparent in our dataset, ore metal budgets play only a secondary role in determining overall ore fertility.

When extended to the full study area, the results of the mantle melting and sulfide fractionation models are also consistent with the observed SMI and whole-rock Cu/Ag behaviour, a proxy for cryptic sulfide saturation[6,10,64,65] (Fig. 6). Sulfide solid:liquid ratios must be taken into account before deriving sulfide fractionation estimates by comparing such chalcophile element ratios in mafic magmas from different settings (e.g., oceanic vs. continental arcs). This is because the incompatibility of Ag relative to Cu (especially in MSS) means that the Cu/Ag ratio in primitive magmas is dependent on the degree of mantle melting and the relative proportions of solid and liquid mantle sulfide. By invoking a mostly solid rather than liquid sulfide in the in the mantle source (the latter invoked for oceanic settings[6]), the modelled Cu/Ag is a good match for those observed in primary SVZ melts and also lower than those of oceanic basalts (e.g., MORB; Fig. 6). This is consistent with the lower mantle melting temperatures at subduction zones due to fluid fluxing. One consequence of the initially lower Cu/Ag ratios is that a lower sulfide fraction is required in the fractionating assemblage to explain the observed relative decreases in Cu/Ag ratios in the SVZ than if starting with MORB-like Cu/Ag. The predicted degree of sulfide fractionation in the lower crust based on the variability of Cu/Ag ratios in the silicate melt (up to $\approx 0.4$ wt%, Fig. 6) is consistent with results of the above calculations based on S mass balance. The constrained amount of sulfide fractionated in the lower crust is especially relevant for models which invoke the remobilization of sulfide-rich reservoirs to generate porphyry-fertile magmas, as the sulfide mass will limit the amount of available chalcophile elements and S.

Taking magmatic sulfide fractionation into account in the north, the observed ore metal concentrations in all magmas presented here can be reproduced by mantle partial melting models which assume an identical mantle source composition half way between those proposed for depleted and primitive mantle[68,69]. Local enrichment or depletion of ore metal concentrations in the mantle source of the magmas are not necessary. However, the elevated S and Cl concentrations in the north likely develop at least partly in response to increased flux of oxidized S and Cl into the mantle wedge by slab-derived fluids[70] and a smaller degree of mantle source melting[32,40]. From this baseline, the volatile budget of northern magmas can further increase during differentiation in manner akin to the 'ramp-up' model of ref. [11].

**General implications for magmatic fertility.** In summary, the southern SVZ is devoid of porphyry Cu deposits because local tectono-magmatic conditions encourage volatile loss during differentiation prior to arrival in the upper crust, further depleting the relatively low initial S and Cl concentrations. The resulting low S and Cl concentrations inhibit efficient ore-metal extraction and transfer to the overlying hydrothermal systems during magma degassing in the upper crust, where a lack of S in the fluid also hinders efficient ore mineral precipitation. The negative impact of this, combined with the relative ease of magma ascent through major crustal structures, outweighs the advantage provided by the relatively high ore metal concentrations in the silicate melt. The northern SVZ generates world class porphyry Cu deposits because the thick crust facilitates the generation of prolonged volatile-rich magmatism where crucial volatile elements such as S and Cl are not significantly depleted in the magma prior to emplacement at upper crustal depths. Sulfur and Cl availability during shallow degassing enable efficient ore metal transport and subsequent ore precipitation, which can mitigate the negative effect of the moderate ore metal scavenging by sulfides during magma differentiation in the lower crust. Increased differentiation within the thicker crust drives the northern magmas toward intermediate-felsic compositions capable of efficient degassing of fluids simultaneously rich in S, Cl, and ore metals after emplacement in upper crustal magma reservoirs.

Overall, our results suggest that the abundance of S and Cl is a more important factor in magmatic ore fertility than the abundance of ore metals themselves. Furthermore, magma differentiation at deeper crustal levels controls ore-fertility not only by sulfide fractionation but also by potential volatile loss via degassing and assimilation-driven dilution. Therefore, it is important that the subducted-slab derived fluids carry significant S and Cl into the mantle wedge, but higher than average initial water concentrations in primitive arc magmas are not favourable for magmatic-hydrothermal ore genesis.

## Methods

### Sample preparation

*Synthetic glass for LA-ICP-MS mass interference tests.* Basaltic andesite glass was synthesized by mixing ultra-pure powder reagents from Alfa Aesar in an agate mortar and pestle under acetone. Aliquots of this powder were separated and doped with one of Zr ($\sim 2000$ ppm), Hf ($\sim 1500$ ppm), or Ta ($\sim 300$ ppm). These powders were then decarbonated in high purity UF4S graphite crucibles at 800 °C for 120 min and then 1000 °C for 30 min during which the crucibles were within a lidded $Al_2O_3$ crucible to prevent oxidation of the graphite and the loss of volatile elements (e.g., Na) from the powder. The graphite crucibles were then transferred to a bottom loading furnace and heated at 1400 °C for 10 min, then rapidly quenched in water creating a glass bead free of quench crystals (except for some rare Fe-oxides). These beads were sectioned through the middle, mounted, and polished.

*Experimental homogenization of crystallized SMI from Maipo.* In contrast to other volcanoes, glassy SMI were very rare or absent in the Maipo samples. To enable the determination of Cl concentrations in SMI from Maipo, we experimentally homogenized originally crystallized SMI to glasses suitable for measurement by EPMA following the method of ref. [71] at 1030 °C and 0.2 GPa for 1.5 h prior to

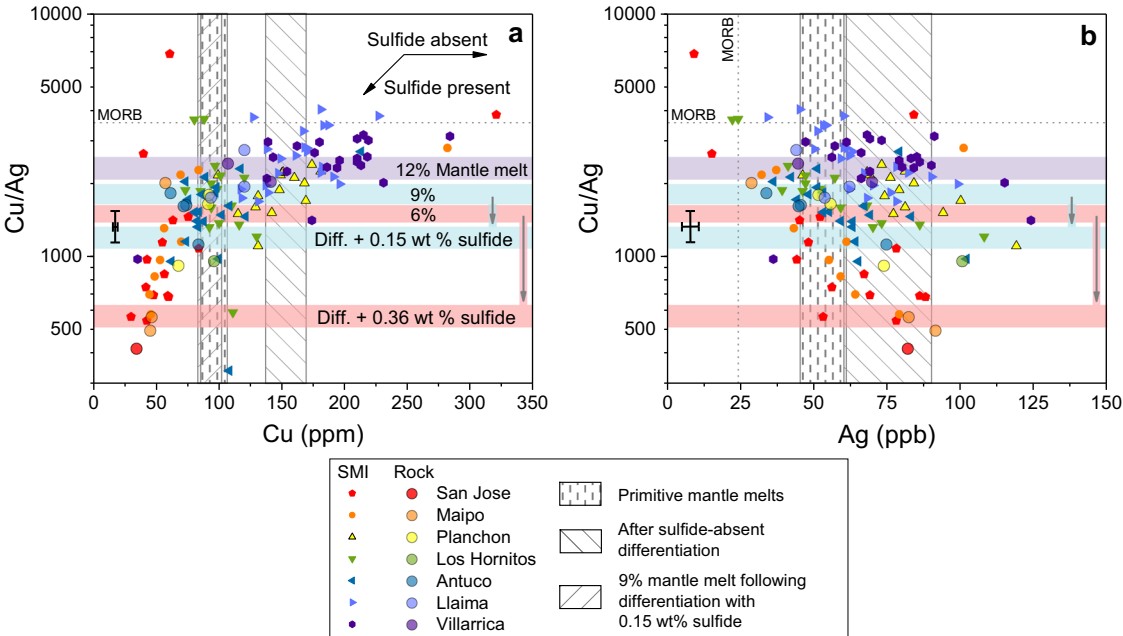

**Fig. 6 Copper/silver ratios in SMI and whole rocks compared to modelled compositions (see 'Methods').** The crosses along the left margin show a representative 1σ uncertainty. Horizontal coloured fields correspond to modelled Cu/Ag ratios at various degrees of mantle melting, whereas vertical fields correspond to Cu or Ag concentrations. Labelled arrows in (**a**) indicate the effect of sulfide saturation during magma differentiation (where differentiation comprises 35% crystal fractionation[66,67]; see 'Methods'). Mantle melting extents range from 12% in the south to 6% in the north[40], and for 9 and 6% melts are paired with differentiation ('Diff.') and sulfide fractionation scenarios described below and in the text. Southern magmas like Villarrica and Llaima did not saturate any significant amount of sulfide and therefore their Cu/Ag ratios match the corresponding mantle melt, even as Cu (**a**) and Ag (**b**) concentrations increase due to the crystallization of major rock forming minerals. Intermediate latitude volcanoes like Antuco and Planchon are on average better represented by moderate melting extents. Although Antuco concentrations and ratios resemble primitive mantle melts, they are consistent with relatively minor sulfide saturation during differentiation[60] (albeit with a slightly lower solid:liquid sulfide ratio than in the north; 0.9:0.1) such that sulfide fractionation offsets the increase of otherwise incompatible Cu. In contrast, the similar mantle melt-like appearance of Los Hornitos likely does reflect the relatively primitive nature of these magmas (host Mg# 83–87, compared to 76–83 for Antuco), which are also likely too oxidized to allow significant sulfide fractionation (Fig. 2). In the north, San Jose and Maipo are consistent with the larger sulfide fractionation estimate described in the text. Silver concentrations (**b**) increase in all volcanoes from mantle melt-like values to those predicted following differentiation to mafic-intermediate compositions.

quenching. Homogenized SMI were identified following sectioning, mounting, and polishing of the heated cores.

**LA-ICP-MS analyses**

*SMI filtering.* Prior to analysis, large SMI (>30–40 μm) were carefully inspected for signs of post-entrapment modification (e.g., decrepitation or pervasive host alteration) and other complications such as co-entrapped phases (typically spinel in olivine-hosted SMI). Only unexposed SMI were selected so that the bulk composition of the silicate melt could be attained even from recrystallized inclusions[72].

*Operating conditions.* Analyses were conducted using an NWR 193 UC laser ablation system coupled to an Agilent 7900 quadrupole inductively coupled plasma mass spectrometer (LA-ICP-MS). Consistent tuning and analytical conditions were used in each session (Supplementary Table 4). Ablation spot dimensions were variable circles and rectangles, depending on the shape and size of each SMI, in order to maximize the relative proportion of the SMI in the mixed SMI-host signal. Whole-rock glasses were analyzed by using a 150 μm beam diameter. Standards were measured with beam sizes matched to the typical size of the SMI in a given analysis batch. If the SMI depth was >50–100% of its diameter, the host was pre-ablated with a large beam size in order to bring the SMI nearer to the surface and minimize elemental fractionation during the SMI analysis.

Whole-rock glasses (Supplementary Fig. 1) were measured in the factory-issued two-volume ablation cell. However, to enable the simultaneous measurement of S with other major and trace elements, the SMI analyses were conducted using a custom low-volume ablation cell characterized by low S backgrounds and a negligible artificial S-signal generated by S remobilization (see also refs. [73,74]). With the exception of San Jose and Maipo, S concentrations determination by LA-ICP-MS and by EMPA from the same samples are consistent with each other (Supplementary Fig. 2) showing that S analysis by LA-ICP-MS are reliable. The good agreement between LA-ICP-MS and EPMA data for S from all other volcanoes suggests that the difference at Maipo and San Jose is not an analytical artefact. As noted in the Fig. 3 caption, we attribute the lower S via EPMA to the fact that the glass phase of the SMI has lost some S to shrinkage bubbles or other S-rich phases that exsolved within the SMI after entrapment[33,34], evidenced by the

systematic localized transient S peak within the LA-ICP-MS signal of the SMI that spreads a significantly narrower time interval than the entire SMI signal (frequently observed at Maipo and San Jose but only rarely at other volcanoes; Supplementary Fig. 5). Such a signal could also be caused by sulfides or anhydrite heterogeneously entrapped with the melt, but this would yield large scatter in the data which has not been observed. Heterogeneous sulfide entrapment would also yield high and scattered Cu concentrations, but this is not apparent either in our dataset (except perhaps for a single pSMI from San Jose with 5500 ppm S and 320 ppm Cu). Therefore, we conclude that some of the S in these inclusions are partitioned into the bubble, and from there may have been subsequently redistributed into S-rich phases like sulfates or Fe-sulfides (explaining the often co-spatial Cu and S peaks). This observation is consistent with the recent study of ref. [33], which identified that >80% of S can be partitioned in the bubble of the SMI during post-entrapment cooling and observed such secondary S-rich phases in the bubble as a result of closed-system SMI behaviour. This highlights the value of LA-ICP-MS analysis of volatile elements, and in particular S in SMI, as this technique always provides the bulk composition of the SMI and not just the composition of the quenched glass phase inside. Note that the degassing of Cl is much less efficient than that of S at the apparent P-T conditions inside the SMI, and therefore the EPMA data for Cl should not be significantly affected by these complications even in the case of the northern volcanoes.

*Data reduction.* Background (~45 s) and signal integration were conducted by using the SILLS software[75] and the mixed SMI-host signals were deconvoluted following the method of ref. [72]. An example SMI analysis is shown in Supplementary Fig. 6. For each SMI, an adjacent measurement of the host was taken using the same beam size as the SMI. The Fe-Mg exchange between olivine and silicate melt ($kD_{olivine-melt}^{Fe-Mg} = 0.31$) was used as an internal standard to deconvolute the mixed SMI-host signals[76]. The limited capacity for post-entrapment olivine crystallization along the walls of these hydrous, mafic-intermediate calc-alkaline melts, should have only a minor effect on the mass factor determination by this method (<3% relative)[76]. For the few pyroxene-hosted SMI, we used correlations between the concentrations and ratios of incompatible major elements (typically CaO, Na$_2$O, and K$_2$O) from olivine-hosted SMI from the same

location[76], or in the case of Maipo, whole-rock $Al_2O_3$ vs. $Na_2O/K_2O$ trends from the data of ref. [77].

*Correction of $^{107}Ag$, $^{197}Au$, and $^{195}Pt$ signals.* Contributions from polyatomic mass interferences to the signals of $^{107}Ag$, $^{197}Au$, and $^{195}Pt$ were corrected based on oxide production rates determined by using mineral standards. These were rich in the interfering element but free of the target element and were measured at both ends of each analysis block to enable the application of drift correction to the oxide production rates (Supplementary Table 5). Signal intervals from all analyses, including standards and unknowns, were corrected for the contribution of oxide interferences before further data processing. These interferences accounted for <30% of the measured signal intensities, as summarized in Supplementary Table 5. The lack of matrix dependence for production rates of mass interferences was confirmed by comparing production rates from the mineral standards to those determined using synthetic basaltic andesite glasses each doped with one of the interfering elements but containing none of the ore metals. The production rates were identical within error in the glasses and minerals, thus only the minerals were used regularly because they provided better counting statistics.

In addition, a Gd-doped glass was synthesized to test argide interferences on $^{195}Pt$ and $^{197}Au$ from $^{155}Gd^{40}Ar$ and $^{157}Gd^{40}Ar$, respectively. However, argide production rates were negligible (i.e. <1% of the oxide production rates) and therefore no correction was applied.

*Calculation of Au and Pt detection limits.* We calculate detection limits following the method of ref. [72], which itself follows ref. [78] for all elements except Au and Pt. Although appropriate for the relatively high count rates typical of most routinely measured elements, common detection limit determinations (e.g., ref. [78]) are inaccurate at low count rates because they have a high probability of erroneously classifying signals as 'detected' (see ref. [79] for a thorough discussion of detection limit determinations). However, the specific version provided by ref. [72] is closely tied to the standard deviation of the background and has the opposite problem. For masses with very low backgrounds (e.g., $^{195}Pt$ and $^{197}Au$) where most sweeps return zero cps but the background has scattered readings of ~10 cps for $^{195}Pt$ and $^{197}Au$ measured with 100 and 120 ms dwell times (see the 0–40 s interval in Supplementary Fig. 6), the standard deviation of the background interval is quite high and the resulting limit of detection is overly conservative.

To address this, ref. [79] provides a formula (their Equation 90) that is better suited for a wide (including very low) range of background intensities and background and signal lengths. Although in this formula $z_\alpha$ is the $z$-value of a given confidence interval, $\alpha$ (e.g., $z_{99\%} = 2.58$), we use an especially cautious value of 3, partly inspired by the dependence on 3 × [background standard deviation] found in the formula of ref. [72] which is used for the rest of the elements. With the use of ref. [79] formula, a detection limit of ~1 ppb was achievable for both Au and Pt in mid to large-sized SMI (e.g., >50 μm).

For all elements, the concentration uncertainties are calculated following the method of ref. [72]. For elements such as Au and Pt with low concentrations, uncertainties are increased by the low count rates and subsequently poor counting statistics.

**EPMA analyses.** Paired analyses of glassy, exposed SMI and their olivine hosts were performed with a JEOL JXA8230 5-WDS electron microprobe. A two-step method was used for SMI. First, major elements were measured with a beam current of 7 nA for 20 s, except Na which was measured for 10 s to mitigate Na migration. Then, S and Cl were measured at 70 nA for 40 s. The beam was defocussed to the maximum size permissible by the exposed SMI surface area (typically 5–10 μm) in order to both maximize the excited volume, and to minimize element migration during analysis.

To resolve the effect of any post-entrapment crystallization of host olivine on the SMI margins, we iteratively added back the composition of the paired host measurement to the SMI until achieving $kD_{olivine-melt}^{Fe-Mg} = 0.31$. The amount of host added back is typically ~1–6% of the total SMI mass.

Due to the scarcity of suitable olivine-hosted SMI in the experimentally homogenized Maipo samples, we instead report the most primitive pyroxene-hosted SMI (~61 wt% $SiO_2$) for EPMA results from Maipo (e.g., Cl concentrations in Figs. 4 and 5).

**Oxygen fugacity estimation and the $S^{2-}$–$S^{6+}$ transition.** We use the V-olivine-melt oxybarometer of ref. [28] because these authors specifically calibrate for hydrous arc basalts, and olivine compositions (Mg# 80–85) similar to our samples. An appropriately calibrated oxybarometer is most important for S, as the pSMI $fO_2$ are within the transition from $S^{2-}$ to $S^{6+}$ predominance (Fig. 2)[29,80,81] which influences the amount of $S^{2-}$ available to potentially saturate an ore metal-scavenging sulfide phase. However, recent studies have proposed that the exact location of the $S^{2-}$–$S^{6+}$ transition will shift to lower $fO_2$ with decreasing pressure[80], but higher $fO_2$ with cooling[29]. Neither of the studies underpinning these scenarios was conducted at the exact conditions invoked in our models. The pressure dependence was investigated at 0.5–1.5 GPa, but lower temperatures (840–950 °C) and with more evolved silicate melt compositions (trachyandesite)[80]. The temperature dependence was investigated through a broader but anhydrous melt composition

range, at atmospheric pressure, only one temperature (1300 °C), and the temperature dependence was addressed by using thermodynamic data rather than direct experiments[29]. The model of ref. [29] does, however, accurately reproduce the results of ref. [80] without invoking pressure.

To avoid uncertainties introduced by the not-yet well quantifiable *P-T* effects on the $fO_2$ range of the $S^{2-}$–$S^{6+}$ transition, we rely on constraints based on well-established sulfide solubility models and measured S concentrations in the SMI to model how magmatic sulfide fractionation affects ore fertility in the SVZ.

### Description of models

*Mantle melting.* The mantle partial melting model assumes non-modal batch melting[82] using the parameters compiled in Supplementary Table 6. Due to debate regarding the degree of enrichment or depletion in the ambient SVZ mantle wedge (e.g., refs. [21,83,84]), we use the average of the otherwise only moderately different chalcophile element concentrations in primitive and depleted mantle model compositions from ref. [69] and ref. [68], respectively. The exception is that the S concentration (220 ppm) is taken to be that of the primitive mantle. This is because a lower sulfide mass proportion, paired with known $S^{2-}$ solubilities and expected degrees of partial melting, would result in (near) total sulfide exhaustion and therefore very high Pt concentrations (e.g., tens of ppb) which are not observed. The employed S concentration is further validated by the agreement of the modelled and measured ore metal concentrations at the expected degree of melting (Fig. 5). One explanation for this comparatively high S concentration in the mantle could be S influx to the mantle-wedge via slab-derived fluids.

The mineral-melt partition coefficients are either experimentally determined or best estimates based on available data. Silicate mineral-melt partition coefficients for Cu are from ref. [85]. Molybdenum partition coefficients for olivine (0.05) and clinopyroxene (0.02) are from SMI-Host pairs in this study. Orthopyroxene is taken to be the average of those two similar values. Molybdenum is incompatible in spinel (e.g., ref. [86]), and based on the behaviour of other HFSE, it is expected to be incompatible in garnet as well[87]. Gold and Ag have similar electronic structures to Cu but larger ionic radii and are therefore treated as even more incompatible than Cu in silicate minerals. This is supported by SMI-host data (e.g., Au and Ag partition coefficients <0.1) and sparse experimental data showing that Au is highly incompatible in olivine and clinopyroxene[88,89]. Knowledge of Pt partitioning between silicate melts and minerals is similarly limited, but Pt has been shown to be incompatible in olivine and spinel[90]. Partition coefficients between monosulfide solid solution (MSS), sulfide liquid (SL) and silicate melt are from ref. [62], except for Pt which is estimated from the 1 atm data of refs. [61,91] using the observation that high-pressure mineral/melt partition coefficients are approximately half the value of those at 1 atm for Cu and Ag[30,62].

*Sulfide fractionation and ore metal scavenging.* To establish a maximum estimate for the amount of fractionated sulfide, we first estimate the maximum available ore metal concentrations. Utilizing our mantle melting model, we estimate the concentration of Cu and Au in mantle melts at appropriate degrees of partial melting in the northern SVZ (~6%)[40]. Even the most primitive basaltic andesitic northern magmas are likely derived from mantle-derived basalts following ~35% crystal fractionation in the lower crust[66,67], so initial ore metal concentrations are then further increased via their incompatible behaviour during the sulfide-absent fraction of silicate minerals. The resulting concentrations are combined with experimentally derived partition coefficients for Cu between the silicate melt and solid sulfide (MORB, 1200 °C, 1.5 GPa)[62] to calculate the amount of sulfide fractionation necessary to yield the measured pSMI Cu concentrations. We calculate that the fractionation of 0.36 wt% sulfide (FeS), equivalent to 1300 ppm $S^{2-}$, would yield ~60 ppm Cu in a basaltic andesite magma that is derived by 35% fractional crystallization of a mantle-derived basalt with ~100 ppm Cu.

We conducted this model calculation assuming a fixed solid to liquid sulfide ratio which was adjusted stepwise until the modelled melt composition matched the measured Cu/Au ratio in pSMI. This method works because the sulfide liquid/pyrrhotite partition coefficient of Au is much higher than that of Cu (Supplementary Table 6). The ratio was found to be 0.97:0.03, validating our initial assumption of dominantly solid sulfide fractionation. The sulfide fractionation estimate is summarized in Supplementary Table 3 and Supplementary Fig. 7.

The basaltic-andesite pSMI compositions from San Jose and Maipo at lower crustal temperatures and pressures (1100 °C; 1–1.5 GPa) and 4.5 wt% $H_2O$ (increased from an initial 3 wt% $H_2O$ following fractionation) yield average $S^{2-}$ concentrations in the silicate melt of ~800 ppm at sulfide saturation (SCSS = sulfur concentration in the silicate melt at sulfide saturation)[92]. Combining the 800 ppm $S^{2-}$ in the pSMI with the 1300 ppm $S^{2-}$ sequestered as sulfide yields a maximum of 2100 ppm $S^{2-}$ in the melt prior to sulfide saturation. The total S concentration measured in the pSMI is up to ~3000 ppm, and so we assume that the remaining 2200 ppm S in excess of the SCSS is present as $S^{6+}$. Combining the $S^{6+}$ and $S^{2-}$ estimates yields a maximum total of 2100 + 2200 = 4300 ppm S (and $S^{2-}/S_{Total}$ of 0.49) in the northern magmas following crystal fractionation but including the precipitated sulfides.

The validity of these estimates can be tested by comparison to both the total S and the $S^{2-}/S_{Total}$ ratio in pSMI from Los Hornitos, which are among the best available approximation of primitive mantle melts in the transitional and northern SVZ. Los Hornitos melts likely undergo very little differentiation as indicated by

olivine compositions (Mg# >85–90; this study and refs. [32,93]) and SMI Cu concentrations that closely match predictions from mantle melting models (~90–100 ppm Cu) prior to sulfide fractionation. Basaltic mantle melts at 1300 °C, 1.5 GPa and with 3 wt% $H_2O$ have an SCSS of ~1300 ppm $S^{2-}$[92]. Therefore, with typical total S concentrations in Los Hornitos SMI reaching ~2500 ppm (this study and ref. [32]) the $S^{2-}/S_{Total}$ ratio is ~0.5. If Los Hornitos melts were to undergo the 35% fractionation inferred for Maipo and San Jose, the total S concentration would reach ~3850 ppm which compares favourably to the above determined value for those volcanoes. The similarity of the total S concentration and the $S^{2-}/S_{Total}$ ratio between the measured primitive Los Hornitos SMI and the calculated primitive northern pSMI validates the model calculations assessing the amount of fractionated sulfide.

The uncertainties on these S concentrations are estimated as follows. The extent of fractional crystallization in the lower crust indicated by refs. [66,67] to derive basaltic andesite from primitive basalt is informed by experiments conducted at lower pressures (0.7 and 1 GPa)[66,67] than we invoke (1–1.5 GPa). Also, our model does not incorporate assimilation-related chalcophile metal contributions during magma differentiation, so we conservatively assume a relative uncertainty of 30% on the estimated hypothetical Cu concentrations before sulfide saturation. Combined with the reported uncertainty for the Cu partition coefficient between silicate melt and solid sulfide (43%)[62] and the standard deviation of northern pSMI Cu concentrations (17%), error propagation yields a 55 relative % (720 ppm) uncertainty on the 1300 ppm $S^{2-}$ fractionated as sulfide. As the 35% crystal fractionation scenario does not account for other potential magma enrichment mechanisms that would make the derivative magmas more felsic but would not increase chalcophile element concentrations as much (e.g., assimilation of crustal material in the lower crust or via subduction erosion), the above numbers for the mass of fractionated sulfide should be considered a maximum estimate.

The 5% uncertainty on $S^{2-}$ solubility[92] combined with the 5% analytical uncertainty on S concentrations in pSMI yields a 7% (56 ppm) uncertainty for the 800 ppm $S^{2-}$ solubility in pSMI melts.

*Constraining deep degassing using S and Cl.* Applying models derived from lower pressures permits proof-of-concept calculations for deep S and Cl degassing scenarios. For example, ref. [41] determined the dependence of Cl partitioning between aqueous fluid and mafic andesite melt (~57 wt% $SiO_2$) from 0.05 to 0.5 GPa. Extrapolating to 1 GPa yields an aqueous fluid-silicate melt partition coefficient of ~57, compared to <5 at 0.2 GPa. Although the degassing behaviour of S at lower crustal pressures is similarly poorly constrained, experiments at upper crustal pressures (0.2 GPa) demonstrate that the partition coefficient of bulk S between fluid and mafic andesite melt decreases from ~175 to ~25 as sulfur transitions from $S^{2-}$ to $S^{6+}$ (ref. [44]). The shape of this partition coefficient curve should not vary with depth, because the shape of the $S^{2-}–S^{6+}$ transition curve is not sensitive to pressure or temperature[29,80].

An open system degassing model using the extrapolated Cl partition coefficient of 57 shows that exsolving ~2 wt% fluid in the lower crust can reproduce the measured decrease in Cl concentrations if initial concentrations are similar to the most primitive Villarrica melts (e.g., ~750–250 ppm Cl; Fig. 4). In contrast, upper crustal degassing (partition coefficient of ~5) will remove only ~30 ppm Cl from the melt, and unreasonable fluid volumes are required to produce the observed 500 ppm decrease (e.g., 24 wt% fluid). The exsolution of 2 wt% fluid can also explain the observed decrease in S concentrations (from ~1500 to ~200 ppm) by using a partition coefficient of ~100. This is a reasonable value assuming both $S^{2-}$ and $S^{6+}$ are present in the melt[44], an assumption supported by the estimated $fO_2$ of these magmas which corresponds to a transitional $S^{6+}$/total S ratio (Fig. 2f).

## Data availability

The data generated in this study are provided in the file Supplementary Data 1 and have also been deposited in the EarthChem Library and can be accessed at https://doi.org/10.26022/IEDA/112330.

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

## Acknowledgements

We thank Alexandra Tsay, Yanan Liu, and Yiwei Yin for analytical assistance and Alice Alex for assistance with experimental work. This project was supported by an NSERC of Canada Discovery Grant to Z.Z. (RGPIN-2014-04805), the Natural Resources Canada - Targeted Geoscience Initiative (Fund #506216 to Z.Z.), generous donations by Dawn Zhou to the research group of ZZ and a Society of Economic Geologists Canada Foundation Student Research Grant to C.G. The infrastructure used in the project was established with the financial support of the Canada Foundation for Innovation – Leaders Opportunity Fund (Grant #32357).

## Author contributions

The project was initially conceived of by Z.Z. C.G. and Z.Z. conducted sample collection and analytical methodology development. C.G. performed sample preparation and analyses and wrote the original draft manuscript, to which Z.Z. provided editing and additional writing.

## Competing interests

The authors declare no competing interests.
