## [Peer Review File · Nature Communications]

Sulfur and chlorine budgets control the ore fertility of arc magmasREVIEWER COMMENTS

Reviewer #1 (Remarks to the Author):

Review of Nature Communications manuscript NCOMMS-20-19898-T entitled "The depth of volatile saturation controls the ore fertility of arc magmas" by Zajacz et al.

In this study, primitive melt inclusions from seven active volcanoes in the Southern Volcanic Zone of the Andes have been analyzed for their content in volatiles and ore-forming elements. The main finding is that primitive melts in the northern, fertile part of the study area contain lower ore metal concentrations but higher volatile concentrations than those erupting in the southern, non-fertile part. The authors conclude that it is the silicate melt's volatile content, rather than its ore-metal content, that determines the potential to produce porphyry-Cu ore deposits. This claim fits well with several recent studies documenting that ore-forming magmas do not contain unusually high metal concentrations (e.g., Zhang and Audétat, 2017; Du and Audétat, 2020), and the current study provides the first direct evidence that it may be the amount of dissolved volatiles that plays the most important role. This finding is novel, of broad interest, and the collected data seem to be of excellent quality (although nowhere tabulated). I thus strongly recommend this work to be eventually published in Nature Communications.

However, there is one issue which I am rather skeptical about: the claim that the southern magmas experienced degassing at mid- to lower-crustal levels. In order to explain the negative correlation between Cl and K₂O at Villarrica the authors extrapolate available fluid–melt partition coefficients to 1 GPa, and they state that exsolution of 2 wt% fluid would explain the decrease in Cl and S. My point is that the H₂O solubility in a basalt at 1 GPa is on the order of 21 wt% (Mitchell et al., 2017), which is 7 times the value of 3 wt% that has been used to model the effect of sulfide precipitation in the northern magmas. I thus severely doubt that aqueous fluid saturation can be attained in primitive magmas at such depths. Even 0.5 GPa, at which pressure about 10 wt% H₂O can be dissolved in basaltic magmas, seems unlikely. Melts containing more than this amount of H₂O cannot be quenched to a glass, hence the presence of glassy melt inclusions points to lower concentrations – independent of whether or not some H₂O was subsequently lost by diffusion. With regard to this I am wondering why no H₂O contents of the melt inclusions are reported. At least some rough estimates based on the deviation of the microprobe totals from 100 wt% should be available.

Given that "The depth of volatile saturation" appears in the title I would expect pressures estimates from mineral thermobarometers to be provided in this work. However, there is not even a description of how many samples have been sampled from each volcano, how these samples compositionally compare to the rest of the volcano, and what their mineralogy and whole-rock composition is. In order to apply thermobarometers, mineral compositions should be provided. Since magmatic sulfides have such a large impact on the availability of chalcophile metals a statement about the presence/absence of magmatic sulfides in samples from the north vs. the south is needed.

Overall, I have the feeling that the significance of the negative S vs. K₂O trends displayed by the melt inclusions from the southern volcanoes is overrated. As explained both in the text and in the supplementary discussion, there is at least one alternative way (other than deep degassing) that could produce these trends. The evidence from Cl vs. K₂O is not quite convincing either, as only Villarrica shows a negative trend, whereas – as mentioned in the supplementary discussion – Llaima and Antuco don't show this (Llaima even displays a positive trend). Without the three negative S vs. K₂O trends and the single negative Cl vs. K₂O trend the story would quite simple: the volatile content of all magmas may originally have been the same, but whereas in the south the magmas are able to ascend relatively easily, in the north they tend to fractionate on the way (perhaps due to the larger compressive stress), with generally more evolved, and thus more volatile-rich (but less metal-rich if sulfide saturation occurred) melts arriving at upper crustal levels, similar to the "ramping up volatiles"-model of Rohrlach and Loucks (2005). At which exact depth some magmas stopped, fractionated, and potentially degassed, would be of course interesting to know, but it is not essential for the main conclusion as long as it can be ruled out that the differences in the volatile and ore metal contents do not arise from processes that operated in upper crustal magma chambers. I thus see the following two options for the revisions: either make a serious effort to constrain pressures through mineralogy/thermobarometry, or to leave it open at which depth level (other than upper crustal level) the differences arise.

Apart from this issue: congratulations on this nice piece of work!

Minor comments keyed to the main text:

line 59: please provide the number of analyses and Tables with all individual LA-ICP-MS and EPMA analyses.

line 66: the Planchon outlier should be mentioned

lines 102-103: there is virtually no evidence provided in the supplementary discussion that would support the statement that "... the characteristics of mafic SVZ magmas are widely accepted to originate in the mantle and/or lower crust".

lines 105-107: I can't quite follow this argument. Who says that the mafic differentiation recorded in individual volcanoes occurred within 104-105 years? Even if true, this time span seems to be easily possible for large, upper crustal magma chambers that are periodically replenished by mafic magmas.

line 117: "geometry, respectively" - both words misspelled

line 159: I would delete the "pyrrhotite or", as pyrrhotite is not stable at magmatic conditions.

lines 189-190: rapid magma ascent and deep degassing somehow does not fit together.

Minor comments keyed to the supplementary discussion:

line 36: Cl correlates negatively with incompatible elements only at Villarrica, whereas Llaima and Antuco do not show such a trend. It is thus not valid to generalize the negative correlation.

lines 54-64: If two out of three trends do not show a negative correlation between Cl and K₂O, then I would not call these trends "anomalous".

Sincerely

References:

Rohrlach B. D. and Loucks R. L. (2005) Multi-million-year cyclic ramp-up of volatiles in a lower crustal magma reservoir trapped below the Tampakan copper-gold deposit by Mio-Pliocene crustal compression in the southern Philippines In Super porphyry copper and gold deposits - A global perspective (ed., T. M. Porter 2, 369-407. PCG Publishing, Adelaide.

Zhang D. and Audétat A. (2017) What caused the formation of the giant Bingham Canyon porphyry Cu-Mo-Au deposit? Insights from melt inclusions and magmatic sulfides. *Econ. Geol.* 112, 221-244.

Du J. and Audétat A. (2020) Early sulphide saturation is not detrimental to porphyry Cu-Au formation. *Geology* 48, <https://doi.org/10.1130/G47169.1>.

Reviewer #2 (Remarks to the Author):

In this study the authors suggest that, based on analysis of melt inclusions, it is the S and Cl budget of magmas that controls the ability to form economic porphyry systems, rather than the concentration of ore-metals in the magma. This, in turn, suggests that if the magmas reach volatile saturation at great depths and lose S and Cl, their ore-forming potential is reduced.

The overall conclusion of this study, that "Overall, our results suggest that the abundance of S and Cl is a more important factor in magmatic ore fertility than the abundance of ore metals themselves." appears to be valid. Other workers have suggested that the efficiency of extraction of metals from the melt may be more important than the actual metal content of the melt, and this work is consistent with those earlier observations and provides a mechanism that relates the extraction efficiency to the Cl and S budget of the melts.

While I have some questions about the details presented in the manuscript, as outlined below, overall the data support the conclusions. This paper is well-organized and well-written, and I recommend acceptance following consideration of the points raised here.

Lines 41-42: please list the host phases for the SMI as this is critical for the reader to understand the details of the analyses. This information is mentioned later and is briefly in the supplementary information, but should be included here.

Line 56: what is meant by "ultra trace level" – can you put a concentration value on this term, or is it just a relative term to imply very low (ppb?) concentrations?

Line 59: did you correct the Fe & Mg concentrations for possible post-entrapment exchange as described by Danyushevsky?

Figure 1: I suggest that you add an "N" at the left end of the X-axis, and an "S" on the right end to clarify "North" and "South" directions.

Lines 64-65: Copper concentrations are distinctly lower in the north than the south (40-75 ppm vs 64 80-200 ppm), ...

One could argue that there really is not much difference in Cu values between the north and south, other than the range is larger in the south. The lower quartile value for Villarica (the southernmost location) is lower than the lowest value reported for the northern locations. I would suggest changing this sentence to:

Average (or median?) copper concentrations, as well as the range in concentrations, increase from north to south

Lines 65-66: and Au and Pt exhibit similar trends though with higher uncertainty due to their ultra-trace 65 level concentrations (1-6 ppb).

Again, I would argue that the trends are weak and if you consider the 1st and 3rd quartiles all of the data overlap.

Lines 136-137: Volatile saturation during 137 magma differentiation at crustal pressures would, however, require particularly water-rich magmas.

To say that this would require water-rich magmas is an under-statement. At 1 GPa, the solubility of water in a basaltic melt is about 20 wt% (Mitchel et al., 2017, Cont. Min. Pet). Stated differently, the melts would contain ~80 mole H₂O and would be better described as aqueous fluids containing some dissolved silicate components. Specifically, elements such as K, Na and others that are transported as chloride complexes might be significantly depleted in the melt as they are lost to an exsolving magmatic fluid. Perhaps a bit more discussion of how you achieve such high water contents to permit volatile saturation at depth is warranted, either here or in the supplemental information.

Lines 141-142: if older plutonic roots are assimilated, might we expect evidence of this in trace element or other chemical signatures?

Reviewer #3 (Remarks to the Author):

To my knowledge this is the first data set containing Cu, Au, Pt, volatiles and trace elements in ol-hosted melt inclusions. The major conclusion is in the last sentence in the abstract. "... the concentration of S and Cl rather than the concentration of the ore metals regulates magmatic hydrothermal ore fertility, and that deep crustal magma degassing reduces ore-forming potential more than magmatic sulfide saturation." This is presently a hotly debated topic and I have some reservations (as discussed below) about the authors' interpretations. This is an important data set that should be published, but the manuscript needs major revision before publication.

It is claimed that very high concentrations of Cl and S in magmas in the northern part of the

Southern Volcanic Zone, where one of the largest porphyry Cu deposits on Earth occurs at Teniente, is the crucial parameter in controlling the formation of porphyry Cu deposits (PCDs). S and Cl are important ligands in ore-forming fluids and thus their high concentration in magma can indeed help to form PCDs. Formation of magmatic sulfide, however, does not necessarily reduce ore-forming potential. It can (and likely does!) work in the opposite manner and actually may be an important prerequisite for PCD formation. Some papers are cited in the manuscript, and there are more recent contributions to this discussion by Du and Audetat (2020, *Geology*), Lee and Tang (2020, *EPSL*), Gilmer et al. (2018, *J. Pet.*), and Cox et al., (2020, *Geology*), which should be taken into account (the latter three papers deal with PCDs in the Andes and the latter two from the SVZ). It is also worth mentioning that early sulfides are not necessarily buried in the lower crust; they can be re-mobilized by fluids and become available for PCD formation (Mungall et al., 2015, *Nature Geosc.*). Therefore, I would encourage the authors to elaborate further on their interpretations and conclusions with respect to newer literature.

More discussion on what controls the S and Cl concentration of the magmas is needed. Why do the authors think that the southern magmas had similarly high S and Cl to those in the north? Could the difference reflect different conditions of magma generation, initially different Cl and S contents? If the positive correlation between K₂O and S remains after correction for fractionation, it would imply a source control, since the magmas more depleted in K₂O have a smaller slab contribution. As noted in the manuscript, there are systematic variations in radiogenic isotope composition along the arc, indicating that magmas consist of different source components and different slab contributions. Thus, why would primitive magmas have uniform volatile contents? The model of deep Cl and S degassing is unclear. Why do magmas in the south degas and those in the north not? Where is the evidence that melt inclusions from the southern volcanoes represent melts from the lower crust, so that the experimental data for high-P fluid-melt partitioning is applicable to them?

The author's also need to include more discussion of published results, in particular of S and Cl in melt inclusions from the Southern Volcanic Zone. Wehrmann et al., 2014 (manuscript reference 41) present an extensive data set on olivine-hosted melt inclusions along the Southern Volcanic Zone from 33 to 43°S. These data suggest that there is not a simple decreasing trend in S going south along the arc from 33-40°S as claimed by the authors. Wehrmann et al. found the highest S contents at Los Hornitos (excluding one analyses from Maipo), which the author's place in their southern group. The aforementioned paper also shows a second, similarly high, peak in S (as well as Cl) at Apagado Volcano (at 42°S). In addition, olivine-hosted melt inclusions from Tupungatito (north of San Jose), despite having very high Cl have very low S. Gilmer et al. (2018) point out variable degassing and decoupling of S and Cl during the complex evolution of magmas in the Pliocene Don Manuel igneous complex ~25 km east of the Teniente PCD. In summary, the volatile systematics appear to be considerably more complex than suggested in the submitted manuscript, raising the question as to how representative the data in the manuscript is. Relevant literature data, esp. from ol-hosted inclusions, needs to be included in figures. Because other results don't agree with the author's data, doesn't mean that they can be ignored.

Along text

Lines 30-39: Some changes are required here to explain potentially important role of early sulfide saturation for PCD formation.

Line 42: Please state which minerals contained the melt inclusions.

Lines 59-61: Some details are required on how this filtering was done (which K₂O content? Why K₂O? Which Fo-number?) and how comparable is the "pSMI" data for different volcanoes?

Lines 63-66: There are some inclusions from the north which have very high Cu (>300 ppm) (fig. 3). What is the origin of these melts? Why not discussed? If these samples are included, the weak inverse correlation between K₂O and Cu disappears. I fail to see that Cu, Au or Pt show systematic trends along the arc (figures 1a-c), as is the case with Mo (fig. 1e). Again, there is the question as to how representative the data set in this manuscript is.

Lines 67-68: The magmas are enriched in Ag compared to MORB and seem to have higher Ag/Cu ratios (not shown). Are there some N-S changes of Ag/Cu ratio and what does it mean in terms of conditions of magma generation in the mantle or magma fractionation in the crust?

Figure 1: What is being plotted here, the measured concentrations or concentrations normalized to a certain K₂O or Fo content?

Figure 2: What does "initial" refer to? Have the concentrations been normalized to a specific K₂O or Fo content? If so what K₂O or Fo content. This needs to be given on the y axis, e.g. S at Fo?.

Literature data need to be included in this figure.

Line 97-98: Fo-number of host olivine is better index than K2O. K2O is not good for comparison of different suites, because the parental melts had different K2O, and K2O could be affected by crustal assimilation (e.g. Hildreth and Moorbath, 1988).

Lines 100-107: Melt inclusions in olivine from volcanic rocks can be trapped at different depths, especially in more evolved whole rocks such as those in the north (in particular the single sample from San Jose). Their compositions do not necessarily represent melts at depth.

Fig. 3: Very weak correlations within individual volcanoes. Are correlations statistically valid?

Line 122: The best argument for more primitive nature of a magma would be high magnesian olivine.

Line 126: Increase in Cu with progressing magma differentiation is only possibly the case for Villarica.

Line 127: High pressure degassing of S and Cl is quite exotic process, and the modeling should involve H₂O and CO₂ components in the magma. The authors could use SolEx program for more realistic modelling (Witham, F., Blundy, J., Kohn, S.C., Lesne, P., Dixon, J., Churakov, S.V., Botcharnikov, R., 2012. SolEx: A model for mixed COHSCI-volatile solubilities and exsolved gas compositions in basalt. *Computers & Geosciences* 45, 87-97).

Geochemically similar effects can also result from low-pressure mixing of primitive and evolved magmas such as in Tolbachik volcano in Kamchatka (e.g. Kamenetsky et al., 2018; *Chem Geology*, Fig. 13).

Line 141: Extensive assimilation is very energy consuming and implies extensive crystallization. There must be some correlations of volatiles and trace elements against Fo-number of olivine. Dzierma et al. (2012) is more appropriate reference here than ref. 33.

Line 157-158: Ideally the authors should demonstrate sulfide inclusions in olivine in the northern magmas. Else this is simply speculation.

Line 158-160: Can it be a primary feature inherited from low-T mantle melting? How can low-T mantle melting be ruled out as the control?

Line 161-181: I think there are many sources of uncertainty involved in this calculation and it is not clear why this estimate is important. Again, the best argument for sulfide crystallization would be photo of sulfide inclusion in primitive olivine.

Line 190-191: Again, I am not convinced that the low Cl and S result from degassing during magma fractionation, instead of from primary magma and/or from low pressure magma mixing.

Line 192-193: "hinders efficient ore mineral precipitation" Sulfide crystallization?

Line 202-207: I partly disagree with this conclusion as explained above.

METHODS:

- It should be explained somewhere that two different datasets are shown on figures: LA-ICP-MS data for unexposed inclusions and 2) EPMA for exposed inclusions. It would be good to demonstrate the consistency of these two data sets for some elements!

- Line 357-360: there is detectable signal of S and K during LA-ICP-MS olivine analysis (ext data Fig.2). Is this real?

- All LA-ICP-MS analyses are done using GSD-1g glass as standard. Are olivine analyses stoichiometric? Can it be demonstrated?

DATA:

There are some problems with Pt - poor data for standards BHVO-2 and TDB-1. The other elements important for this work - Au, Ag and Cu seem to be ~ Ok.

The source of reference data for TDB-1 should be checked; most elements were not reported in the cited source.

Sc and Cu data in olivine should be corrected for interference with SiO on mass 45Sc and MgAr on mass 65Cu. If not corrected then it must be indicated or data excluded.

Much data on olivine host have elevated Ba (above DL), which correlates with Na, K, Sr, LREE and other highly incompatible in olivine elements and indicates contamination by LILE-LREE rich glass. All data must be checked before publication, and analyses affected by the contamination (Ba above detection limit) should be clearly indicated and affected elements excluded.

**NCOMMS-20-19898-T** (by Grondahl and Zajacz)

**RESPONSE TO REVIEWER COMMENTS**

Please see our responses in blue following each comment of the reviewers.

**Reviewer #1 (Remarks to the Author):**

Review of Nature Communications manuscript NCOMMS-20-19898-T entitled "The depth of volatile
saturation controls the ore fertility of arc magmas" by Zajacz et al.

In this study, primitive melt inclusions from seven active volcanoes in the Southern Volcanic Zone of
the Andes have been analyzed for their content in volatiles and ore-forming elements. The main finding
is that primitive melts in the northern, fertile part of the study area contain lower ore metal
concentrations but higher volatile concentrations than those erupting in the southern, non-fertile part.
The authors conclude that it is the silicate melt's volatile content, rather than its ore-metal content, that
determines the potential to produce porphyry-Cu ore deposits. This claim fits well with several recent
studies documenting that ore-forming magmas do not contain unusually high metal concentrations (e.g.,
Zhang and Audétat, 2017; Du and Audétat, 2020), and the current study provides the first direct
evidence that it may be the amount of dissolved volatiles that plays the most important role. This
finding is novel, of broad interest, and the collected data seem to be of excellent quality (although
nowhere tabulated). I thus strongly recommend this work to be eventually published in Nature
Communications.

However, there is one issue which I am rather skeptical about: the claim that the southern magmas
experienced degassing at mid- to lower-crustal levels. In order to explain the negative correlation
between Cl and K₂O at Villarrica the authors extrapolate available fluid–melt partition coefficients to 1
27 GPa, and they state that exsolution of 2 wt% fluid would explain the decrease in Cl and S. My point is
28 that the H₂O solubility in a basalt at 1 GPa is on the order of 21 wt% (Mitchell et al., 2017), which is 7
29 times the value of 3 wt% that has been used to model the effect of sulfide precipitation in the northern
magmas. I thus severely doubt that aqueous fluid saturation can be attained in primitive magmas at
such depths. Even 0.5 GPa, at which pressure about 10 wt% H₂O can be dissolved in basaltic magmas,
seems unlikely. Melts containing more than this amount of H₂O cannot be quenched to a glass, hence
the presence of glassy melt inclusions points to lower concentrations – independent of whether or not
some H₂O was subsequently lost by diffusion. With regard to this I am wondering why no H₂O contents
of the melt inclusions are reported. At least some rough estimates based on the deviation of the
microprobe totals from 100 wt% should be available.

The challenge of attaining fluid saturation at greater than upper crustal depths is now discussed
in the main text at lines 177-192. We note that the degassing of aqueous fluids will be facilitated to
some extent by the much lower (and also pressure dependent) solubility of CO₂, which will
simultaneously remove H₂O from the melt upon saturation. This is an important aspect for which we
have added some additional supporting argumentation via MELTS modelling of C-O-H fluid saturation
(lines 182-185), but there is a lack of proper experimental constraints needed to model degassing of Cl-
bearing fluids from mafic systems at mid to lower crustal depth. However, the point stands that

experiments indicate that shallow degassing is an inefficient way to remove Cl from the melt, and Cl
partitions into the fluid more strongly as pressure increases.

Importantly, we think that the studied silicate melt inclusions (SMI) were trapped at upper
crustal depths during early phenocryst growth and represent melts from different batches of magma
which fed the upper crustal magma reservoirs and mixed with each other at that level. The
compositional variation was, however, imposed by deeper processes, which affected the composition of
the magma batches before their arrival in the upper crust. This point is now stated explicitly at line 47-
49 (see also Kent, 2008, RiMG v69, Gurenko et al., 2005, JPet, and Humphreys et al., 2008, Contrib Min
Pet). Therefore, the H₂O concentrations in the SMI were likely equilibrated at the $f_{\text{H}_2\text{O}}$ apparent in the
upper crustal magma reservoir via diffusion (e.g., Gaetani et al., 2012, Geology) and then were further
perturbed during syn- and post-eruptive cooling and degassing. Original EPMA totals are given in the
Supplementary Data, but throughout the SVZ deviate by less than 2 wt% from 100 wt%, and thus we
expect that the water concentrations do not provide quantitative insights to magmatic H₂O budgets for
the reasons stated above.

Given that "The depth of volatile saturation" appears in the title I would expect pressures estimates
from mineral thermobarometers to be provided in this work. However, there is not even a description of
how many samples have been sampled from each volcano, how these samples compositionally compare
to the rest of the volcano, and what their mineralogy and whole-rock composition is. In order to apply
thermobarometers, mineral compositions should be provided. Since magmatic sulfides have such a large
impact on the availability of chalcophile metals a statement about the presence/absence of magmatic
sulfides in samples from the north vs. the south is needed.

The number of samples, their whole rock compositions and the composition of the host
minerals are provided in the Supplementary Data, and the number of samples are now mentioned at
line 54 in the main text. Additionally, the compositions are summarized in the context of literature data
for each volcano in new Extended Data Figure 2. A statement has been added (line 83) noting the rarity
and/or absence of magmatic sulfides in all samples (this is also tabulated in a new Extended Data Table
2), emphasizing that observed S and chalcophile element behaviour is not derived from upper crustal
magmatic sulfide saturation.

Specific depth determinations and thermobarometry were not undertaken in this study because
the observed phase assemblages (now tabulated in the Supplementary Data) indicate predominantly
upper crustal crystallization, and within the relevant pressure range the relative error of these methods
is too high for pressure estimation. We now clarify the point at lines 46-53 such that we do not imply
that the melt inclusions were entrapped at lower crustal pressures. Also, please note that most SMI
were olivine hosted, and therefore we cannot obtain pressure estimates directly from the host mineral
compositions.

Overall, I have the feeling that the significance of the negative S vs. K₂O trends displayed by the
melt inclusions from the southern volcanoes is overrated. As explained both in the text and in the
supplementary discussion, there is at least one alternative way (other than deep degassing) that could
produce these trends. The evidence from Cl vs. K₂O is not quite convincing either, as only Villarrica
shows a negative trend, whereas – as mentioned in the supplementary discussion – Llaima and Antuco
don't show this (Llaima even displays a positive trend). Without the three negative S vs. K₂O trends and

the single negative Cl vs. K₂O trend the story would quite simple: the volatile content of all magmas may
originally have been the same, but whereas in the south the magmas are able to ascend relatively easily,
in the north they tend to fractionate on the way (perhaps due to the larger compressive stress), with
generally more evolved, and thus more volatile-rich (but less metal-rich if sulfide saturation occurred)
melts arriving at upper crustal levels, similar to the "ramping up volatiles"-model of Rohrlach and Loucks
(2005). At which exact depth some magmas stopped, fractionated, and potentially degassed, would be
of course interesting to know, but it is not essential for the main conclusion as long as it can be ruled out
that the differences in the volatile and ore metal contents do not arise from processes that operated in
upper crustal magma chambers. I thus see the following two options for the revisions: either make a
serious effort to constrain pressures through mineralogy/thermobarometry, or to leave it open at which
depth level (other than upper crustal level) the differences arise.

As suggested, we have rephrased some arguments such that an upper-crustal origin is still ruled
out, but otherwise the depth at which the observed trends developed is left open. With this point in
mind, we have softened wording that explicitly invokes the lower crust as the site of degassing-related S
and Cl loss in the southern magmas in general, and propose Villarrica as an end-member scenario from
which deviations will arise via local tectono-magmatic circumstances as at Antuco and Llama. Though
we still emphasize experimental constraints suggesting that the degassing processes which played role
in the generation of the observed trends must have happened at greater than upper crustal depths, we
also discuss in more detail alternative processes that may explain the observed trends (e.g. dilution by
assimilation in the lower crust, source control). Furthermore, as suggested by the reviewer, we compare
our results favourably to those of Rohrlach and Loucks (2005), but importantly at an arc-wide scale
rather than in a single well-studied system. We would like to note though that the degree of
differentiation is non-negligible in the southern magmas, though smaller than it is in the north.

Apart from this issue: congratulations on this nice piece of work!

Minor comments keyed to the main text:

line **59**: please provide the number of analyses and Tables with all individual LA-ICP-MS and EPMA
analyses.

Number of analyses now specified for pSMI and fSMI, and these Tables are provided in the
Supplementary Data.

line **66**: the Planchon outlier should be mentioned

This section is now worded that the range in Cu concentrations is greater south of Maipo, which
is less strict phrasing, and Planchon is specifically mentioned in the Fig. 1 caption.

lines **102-103**: there is virtually no evidence provided in the supplementary discussion that would
support the statement that "... the characteristics of mafic SVZ magmas are widely accepted to
originate in the mantle and/or lower crust".

We have added a brief comment to the Supplementary Text section 'Linking the modern arc to
past porphyry Cu ore genesis' (lines 12-13) which specifically points out that the various summarized

models all invoke processes operating at depths of at least the mid-lower crust, and changed the line
(123) in the main text to say that the characteristics originate “prior to arrival in the upper crust”.

lines **105-107**: I can't quite follow this argument. Who says that the mafic differentiation recorded in
individual volcanoes occurred within 104-105 years? Even if true, this time span seems to be easily
possible for large, upper crustal magma chambers that are periodically replenished by mafic magmas.

This point has now been generalized, citing the work of Annen et al. (2006, JPet.), to the effect
that the large volume of similar-composition magmas typical of mature arc settings (evidenced by
plutonic and volcanic products) are a consequence of long-lived, relatively stable magma storage which
itself implies a hotter geotherm.

line **117**: "geometry, respectively" - both words misspelled

Corrected.

line **159**: I would delete the "pyrrhotite or", as pyrrhotite is not stable at magmatic conditions.

Implemented.

lines **189-190**: rapid magma ascent and deep degassing somehow does not fit together.

Rephrased to say that “local tectono-magmatic conditions encourage volatile loss during
differentiation prior to arrival in the upper crust, further depleting the relatively low initial S and Cl
concentrations”. Provided that the volatile content is sufficiently high to reach volatile saturation at
greater depths, decompression-driven degassing of mafic magmas is a relatively fast process in
comparison to crystallization.

Minor comments keyed to the supplementary discussion:

line **36**: Cl correlates negatively with incompatible elements only at Villarrica, whereas Llaima and
Antuco do not show such a trend. It is thus not valid to generalize the negative correlation.

We no longer make this generalization.

lines **54-64**: If two out of three trends do not show a negative correlation between Cl and K₂O, then I
would not call these trends "anomalous".

These are now discussed in the main text, and no longer presented as anomalies. We frame
Llaima and Antuco, citing other studies on each, as local variations on a Villarrica-like end-member that
have seen greater degrees of shallow pre-eruptive differentiation and/or mixing.

Sincerely

References:

Rohrlach B. D. and Loucks R. L. (2005) Multi-million-year cyclic ramp-up of volatiles in a lower crustal
magma reservoir trapped below the Tampakan copper-gold deposit by Mio-Pliocene crustal

compression in the southern Philippines In Super porphyry copper and gold deposits - A global
perspective (ed., T. M. Porter 2, 369-407. PCG Publishing, Adelaide.

Zhang D. and Audétat A. (2017) What caused the formation of the giant Bingham Canyon porphyry Cu-
Mo-Au deposit? Insights from melt inclusions and magmatic sulfides. Econ. Geol. 112, 221-244.

Du J. and Audétat A. (2020) Early sulphide saturation is not detrimental to porphyry Cu-Au formation.
Geology 48, <https://doi.org/10.1130/G47169.1>.

**Reviewer #2 (Remarks to the Author):**

In this study the authors suggest that, based on analysis of melt inclusions, it is the S and Cl budget of
magmas that controls the ability to form economic porphyry systems, rather than the concentration of
ore-metals in the magma. This, in turn, suggests that if the magmas reach volatile saturation at great
depths and lose S and Cl, their ore-forming potential is reduced.

The overall conclusion of this study, that “Overall, our results suggest that the abundance of S and Cl is a
more important factor in magmatic ore fertility than the abundance of ore metals themselves.” appears
to be valid. Other workers have suggested that the efficiency of extraction of metals from the melt may
be more important than the actual metal content of the melt, and this work is consistent with those
earlier observations and provides a mechanism that relates the extraction efficiency to the Cl and S
budget of the melts.

While I have some questions about the details presented in the manuscript, as outlined below, overall
the data support the conclusions. This paper is well-organized and well-written, and I recommend
acceptance following consideration of the points raised here.

Lines **41-42**: please list the host phases for the SMI as this is critical for the reader to understand the
details of the analyses. This information is mentioned later and is briefly in the supplementary
information, but should be included here.

We now specify here that the host minerals are olivine and pyroxene, and have also added the
number of SMI that have olivine, clinopyroxene, or orthopyroxene hosts (and host Mg# ranges) in the
next paragraph (lines 68-70). Additionally, the host of each SMI measured by LA-ICP-MS is now named in
the Supplemental Data (SMI measured by EPMA are all olivine-hosted).

Line **56**: what is meant by “ultra trace level” – can you put a concentration value on this term, or is it just
a relative term to imply very low (ppb?) concentrations?

The latter- this is now explicitly stated as <0.1 ppm.

Line 59: did you correct the Fe & Mg concentrations for possible post-entrapment exchange as
described by Danyushevsky?

We did not do such a correction. As discussed by Zajacz and Halter (2007; GCA), this will likely
have a very minor effect on the mass factor determination (<3% relative) because of the limited capacity
for the SMI to crystallize further olivine on the inclusion walls after entrapment, which in turn could re-
equilibrate with the host during cooling. This is a characteristic of hydrous, calc-alkaline basaltic andesite
melts as opposed to MOR basalts, in which olivine-hosted SMI are capable of crystallizing large amounts
of olivine on the walls of the inclusion. This is now noted in the Methods at lines 559-561.

**Figure 1:** I suggest that you add an “N” at the left end of the X-axis, and an “S” on the right end to clarify
“North” and “South” directions.

Implemented.

Lines 64-65: Copper concentrations are distinctly lower in the north than the south (40-75 ppm vs 64 80-
200 ppm),

One could argue that there really is not much difference in Cu values between the north and south,
other than the range is larger in the south. The lower quartile value for Villarica (the southernmost
location) is lower than the lowest value reported for the northern locations. I would suggest changing
this sentence to:

Average (or median?) copper concentrations, as well as the range in concentrations, increase from north
to south

Implemented as suggested.

Lines 65-66: and Au and Pt exhibit similar trends though with higher uncertainty due to their ultra-
trace 65 level concentrations (1-6 ppb).

Again, I would argue that the trends are weak and if you consider the 1st and 3rd quartiles all of the
data overlap.

We now state that clear trends in Au and Pt are obscured by the high analytical uncertainties.

Lines 136-137: Volatile saturation during 137 magma differentiation at crustal pressures would,
however, require particularly water-rich magmas.

To say that this would require water-rich magmas is an under-statement. At 1 GPa, the solubility of
water in a basaltic melt is about 20 wt% (Mitchel et al., 2017, Cont. Min. Pet). Stated differently, the
melts would contain ~80 mole H₂O and would be better described as aqueous fluids containing some
dissolved silicate components. Specifically, elements such as K, Na and others that are transported as
chloride complexes might be significantly depleted in the melt as they are lost to an exsolving magmatic
fluid. Perhaps a bit more discussion of how you achieve such high water contents to permit volatile
saturation at depth is warranted, either here or in the supplemental information.

We no longer specifically invoke lower crustal degassing. However, we discuss the difficulty of
depleting Cl in mafic magmas at shallow crustal depth and argue for degassing that is at least initiated
prior to arrival in the upper crust. This could happen at mid-crustal levels where water saturation can be
achieved at 8-10 wt% H₂O concentrations even in the absence of CO₂. Primitive arc magmas may contain
up to >6 wt% H₂O in some cases (Grove et al., 2012, AnRevEPS). These H₂O concentrations could be
elevated to saturation levels in the mid-crust by degrees of crystallization that allow for the observed

basaltic-andesite derivative melt composition. This is now discussed in detail at lines 177-192.

Lines **141-142**: if older plutonic roots are assimilated, might we expect evidence of this in trace element
or other chemical signatures?

Yes, and this is most readily seen as an increase of incompatible element concentrations that
outpaces what could be attributed to fractional crystallization over a limited observed range of major
element concentrations (e.g., Hildreth and Moorbath, 1988, *ConMinPet*; DePaolo, 1981, *EPSL*). In the
likely case that the plutonic roots have similar isotopic compositions to the modern magma, the
traditional isotopic evidence will be muted (e.g., Dungan and Davidson, 2004, *Geology*, Reubi et al.,
2011, *EPSL*). Extended Data Figures 4 and 5 summarize this for our samples.

**Reviewer #3 (Remarks to the Author):**

To my knowledge this is the first data set containing Cu, Au, Pt, volatiles and trace elements in ol-hosted
melt inclusions. The major conclusion is in the last sentence in the abstract. "... the concentration of S
and Cl rather than the concentration of the ore metals regulates magmatic hydrothermal ore fertility,
and that deep crustal magma degassing reduces ore-forming potential more than magmatic sulfide
saturation." This is presently a hotly debated topic and I have some reservations (as discussed below)
about the authors' interpretations. This is an important data set that should be published, but the
manuscript needs major revision before publication.

It is claimed that very high concentrations of Cl and S in magmas in the northern part of the Southern
Volcanic Zone, where one of the largest porphyry Cu deposits on Earth occurs at Teniente, is the crucial
parameter in controlling the formation of porphyry Cu deposits (PCDs). S and Cl are important ligands in
ore-forming fluids and thus their high concentration in magma can indeed help to form PCDs. Formation
of magmatic sulfide, however, does not necessarily reduce ore-forming potential. It can (and likely
does!) work in the opposite manner and actually may be an important prerequisite for PCD formation.
Some papers are cited in the manuscript, and there are more recent contributions to this discussion by
Du and Audetat (2020, *Geology*), Lee and Tang (2020, *EPSL*), Gilmer et al. (2018, *J. Pet.*), and Cox et al.,
(2020, *Geology*), which should be taken into account (the latter three papers deal with PCDs in the
Andes and the latter two from the SVZ). It is also worth mentioning that early sulfides are not
necessarily buried in the lower crust; they can be re-mobilized by fluids and become available for PCD
formation (Mungall et al., 2015, *Nature Geosc*). Therefore, I would encourage the authors to elaborate
further on their interpretations and conclusions with respect to newer literature.

It is intuitive that higher ore metal concentrations in the magma should result in higher ore metal
concentrations in the fluid, and therefore lead to the generation of more metal rich hydrothermal fluids
during upper-crustal magma degassing. However, in part due to rarity of SMI data from porphyry Cu-
ore deposit related magmas, it remains to be determined what the ore metal, S and Cl budget in a fertile
magma actually is. Put another way, the expected capacity of sulfides to remove Cu from the melt may
not on its own be enough to dictate the fertility of potential porphyry-forming magmas (here we've
argued that efficient ore metal transport and extraction via S- and Cl-bearing fluids is the general
limiting factor on the magmatic side). A subsequent step following sulfide saturation (e.g., sulfide
breakdown, vapour transport..) is always invoked in order to involve sulfide saturation as a key process

increasing magmatic fertility, and therefore it is currently a matter of opinion whether this explains the
scarcity of porphyry deposits or is instead too delicate of a balance to invoke in most porphyry deposit-
forming magmatic systems. In our opinion, the remobilization of sulfides and their metal budget from
lower crustal cumulates may be challenging, but the detailed discussion of this goes beyond the scope of
the present manuscript.

Regarding the study of Mungall et al. (2015, NatGeo), these authors' experiments and their
modeling exercises addressed upper crustal conditions. Although the mechanism could perhaps be
extended explicitly to the lower crust, we do not see evidence of extensive degassing in the northern
lower crust where sulfides are inferred to be an influential phase, and in fact limited degassing in the
lower crust is likely an important attribute of fertile systems (e.g., Rohrlach and Loucks, 2005).
Southward through the SVZ, significant Cu depletion via sulfide fractionation does not seem to occur
and therefore ore metals would not be greatly mobilized in this manner whenever (and wherever)
degassing occurs.

Thank you for drawing our attention to the above-mentioned recent studies, which have now
been taken into consideration and cited where appropriate within the manuscript.

More discussion on what controls the S and Cl concentration of the magmas is needed. Why do the
authors think that the southern magmas had similarly high S and Cl to those in the north? Could the
difference reflect different conditions of magma generation, initially different Cl and S contents? If the
positive correlation between K₂O and S remains after correction for fractionation, it would imply a
source control, since the magmas more depleted in K₂O have a smaller slab contribution. As noted in
the manuscript, there are systematic variations in radiogenic isotope composition along the arc,
indicating that magmas consist of different source components and different slab contributions. Thus,
why would primitive magmas have uniform volatile contents?

We do not claim that primitive magmas throughout the SVZ have uniform volatile contents and in
fact propose the opposite. This is stated specifically via pSMI in main text lines 84-86, and shown in the
data of Figures 2 and 3. We also have added a sentence at lines 211-212 which cites Wehrmann et al.
(2014, IJEarthSci)'s conclusion that the Cl endowment in the initial mantle melts is at least partly related
to the degree of source melting. We now more specifically attribute intra-volcano variations of volatile
element concentrations to local circumstances affecting the depth of storage, rate of ascent,
composition, etc., and do not imply a smooth, uniform evolution from north to south. The difference in
factors affecting intra- vs. inter-volcano variations in volatile budgets and behaviours is summarized in
lines 211-222.

The model of deep Cl and S degassing is unclear. Why do magmas in the south degas and those in the
north not? Where is the evidence that melt inclusions from the southern volcanoes represent melts
from the lower crust, so that the experimental data for high-P fluid-melt partitioning is applicable to
them?

The ease of degassing is likely related to the contrast in crustal permeability implied by the major
Loquiñe-Ofqui Fault Zone occurring south of ~Antuco, and will be compounded by local variability in
volatile (CO₂ and H₂O) budgets. The (p)SMI represent the closest available approximation of lower
crustal melts because they preserve a record of various magma batches that equilibrated at depth and
amalgamated in the upper crust prior to eruption of the magmas generating volcanic rocks within which
these various magma histories are averaged out; see Kent, 2008, RiMG v69). We thus use the SMI data

in calculations to test our conceptual models, but do not argue that the SMI are exact equivalents of
lower crustal melts (nor that they were trapped in the lower crust). This is now clarified specifically at
lines 47-53.

The author's also need to include more discussion of published results, in particular of S and Cl in melt
inclusions from the Southern Volcanic Zone. Wehrmann et al., 2014 (manuscript reference 41) present
an extensive data set on olivine-hosted melt inclusions along the Southern Volcanic Zone from 33 to
43°S. These data suggest that there is not a simple decreasing trend in S going south along the arc from
33-40°S as claimed by the authors. Wehrmann et al. found the highest S contents at Los Hornitos
(excluding one analyses from Maipo), which the author's place in their southern group. The
aforementioned paper also shows a second, similarly high, peak in S (as well as Cl) at Apagado Volcano
(at 42°S). In addition, olivine-hosted melt inclusions from Tupungatito (north of San Jose), despite having
very high Cl have very low S. Gilmer et al. (2018) point out variable degassing and decoupling of S and Cl
during the complex evolution of magmas in the Pliocene Don Manuel igneous complex ~25 km east of
the Teniente PCD. In summary, the volatile systematics appear to be considerably more complex than
suggested in the submitted manuscript, raising the question as to how representative the data in the
manuscript is. Relevant literature data, esp. from ol-hosted inclusions, needs to be included in figures.
Because other results don't agree with the author's data, doesn't mean that they can be ignored.

The data of Wehrmann et al. (2014), which the authors first filter for the 'least-degassed' SMI, is
in good agreement with our SMI data for the volcanoes we studied and we now state this explicitly in
the Figure 2 caption. The exception, as we now note, is that Maipo and San Jose have somewhat higher
S concentrations in LA-ICP-MS data than in EPMA data. Chlorine, not measured by LA-ICP-MS, agrees
well between our and the Wehrmann dataset. The S discrepancy is attributed to post-entrapment
migration of S from the melt to the bubble phase within the SMI (e.g. Moore et al. 2015 - Am Min and
Venugopal et al. 2020 - Nature Sci Rep), and highlights the importance of bulk SMI measurement
techniques such as LA-ICP-MS which, unlike EPMA, do not exclude phases within the SMI that could act
as important reservoirs of key elements (note that bubbles in SMI most often cannot be re-dissolved
even during heating experiments). A thorough discussion of this has been added to the Methods (lines
530-552).

This argumentation may also partly explain the lower S at Tupungatito in the Wehrmann et al.
study, although here shallow degassing prior to SMI formation may also be important because the S
concentrations (<500 ppm) and S/Cl are rather low in the SMI. Furthermore, only three SMI were
reported from Tupungatito and therefore additional data are needed for a more robust interpretation of
that system.

We also note that Gilmer et al. (2018) have inferred their S and Cl inventories in early basaltic
andesite melts on an indirect manner and qualitative basis only, based largely on S and Cl in apatite with
considerable scatter in the data (their Fig. 11). Inferring melt compositions from apatite compositions is
currently somewhat ambiguous due the lack of appropriate apatite/melt partitioning models accounting
for the effect of melt composition, *P* and *T*. In addition, we are hesitant to extend their conclusions to
deep, early magma conditions (like those potentially available via SMI analyses) because of their reliance
on apatite from basaltic andesite dikes for their mafic-intermediate data. These apatites are small,
~acicular, and may well have formed after or concurrent with significant degassing during dike
emplacement as noted by the authors and evidenced by their very low S/Cl ratios. Note that apatite
saturation generally requires relatively low temperatures.

Along text

Lines **30-39**: Some changes are required here to explain potentially important role of early sulfide
saturation for PCD formation.

We feel this paragraph concisely outlines the proposed importance of this process. Namely, that
sulfide-related accumulations of S and ore metals can potentially be later released simultaneously to
generate especially S- and ore metal-rich magma batches.

Line **42**: Please state which minerals contained the melt inclusions.

We now state here that the SMI are olivine- and pyroxene-hosted, and provide the number of SMI
in each host (and host Mg#'s) in the following paragraph (line 68-70).

Lines **59-61**: Some details are required on how this filtering was done (which K2O content? Why K2O?
Which Fo-number?) and how comparable is the "pSMI" data for different volcanoes?

A new Extended Data Table (Table 1) has been added which details the filtering criteria for each
volcano.

Lines **63-66**: There are some inclusions from the north which have very high Cu (>300 ppm) (fig. 3). What
is the origin of these melts? Why not discussed? If these samples are included, the weak inverse
correlation between K2O and Cu disappears. I fail to see that Cu, Au or Pt show systematic trends along
the arc (**figures 1a-c**), as is the case with Mo (**fig. 1e**). Again, there is the question as to how
representative the data set in this manuscript is.

The question regarding clarity of trends in the pSMI data of Figure 1 has been addressed
(following another Reviewer's suggestions) such that we acknowledge greater southward variability in
Cu concentrations as well as state less-firmly that Au and Pt trends are clearly discernable.

The high-Cu (>100 ppm) inclusions from the north include three SMI from San Jose and one from
Maipo. Anomalously high Cu concentrations could hypothetically be attributed to heterogeneous
entrapment of silicate melt and magmatic sulfide. That scenario seems clear only for the one SMI from
San Jose (321 ppm Cu, 5539 ppm S). Another possibility is that these SMI are relicts of magma batches
which for some reason avoided sulfide fractionation during differentiation, reminiscent of Villarrica and
Llaima. In any case, these four high-Cu SMI are a rare exception to the rule of fairly uniform S-Cu
concentrations in the northern SMI, and this uniformity is evidence in favour of the representativeness
of the SMI compositions.

Lines **67-68**: The magmas are enriched in Ag compared to MORB and seem to have higher Ag/Cu
ratios (not shown). Are there some N-S changes of Ag/Cu ratio and what does it mean in terms of
conditions of magma generation in the mantle or magma fractionation in the crust?

We have dedicated a new figure (5) and text (lines 249-263) to Cu/Ag ratios from both SMI and
whole rocks. We show that our mantle melting and sulfide fractionation models (and their broad
conclusions) also explain the observed Cu/Ag behaviour, and compare the implications of our models to
those proposed for oceanic basalts (Jenner, 2017, NatGeo).

**Figure 1**: What is being plotted here, the measured concentrations or concentrations normalized to a
certain K2O or Fo content?

This is pSMI data, which are subgroups from each volcano filtered based on K₂O and Fo, and are
not normalized. All data presented is unnormalized, except where specifically stated.

**Figure 2:** What does “initial” refer to? Have the concentrations been normalized to a specific K₂O or Fo
content? If so what K₂O or Fo content. This needs to be given on the y axis, e.g. S at Fo?. Literature data
need to be included in this figure.

‘Initial’ has been rephrased as ‘relatively primitive’. These are not normalized explicitly, but as
indicated, are the pSMI data (filtered for high Fo content in the host and low K₂O in the SMI; the filtering
criteria are in new Extended Data Table 1). This is already a somewhat crowded figure, and so a detailed
comparison of S and Cl concentration between this study and that of Wehrmann et al. (2014, IJEarthSci)
is not presented. However, we point out the consistency between our and their data in the caption.

Line **97-98:** Fo-number of host olivine is better index than K₂O. K₂O is not good for comparison of
different suites, because the parental melts had different K₂O, and K₂O could be affected by crustal
assimilation (e.g. Hildreth and Moorbath, 1988).

We would argue that K₂O is useful for the very reason that variable crustal assimilation occurs and
K₂O will be sensitive to this also, rather than more dominantly reflecting crystal fractionation alone as
would be the case for the host olivine Fo-numbers. Thus using K₂O may offer a more comprehensive
look at differentiation. In addition, the Fe/Mg ratios in olivine can rapidly re-equilibrate with the residual
melt after crystallization, and this will likely be a prominent process during magma differentiation in the
mid to lower crust which happens on much longer timescales than upper crustal magma differentiation.
In contrast, K diffusion in olivine is very slow.

Furthermore, in this figure we specifically emphasize the contrasting behaviour of S and Cl as
differentiation proceeds (e.g., K₂O increases) at each volcano.

Lines **100-107:** Melt inclusions in olivine from volcanic rocks can be trapped at different depths,
especially in more evolved whole rocks such as those in the north (in particular the single sample from
San Jose). Their compositions do not necessarily represent melts at depth.

We do not claim that they are direct measurements of melts existing at lower crustal conditions,
merely that they represent the closest available approximation of such melts, which have since
ascended and been trapped as SMI in the upper crust.

**Fig. 3:** Very weak correlations within individual volcanoes. Are correlations statistically valid?

The text has been altered to no longer imply the generalization that all southern magmas have
clear trends and correlations like Villarica.

Line **122:** The best argument for more primitive nature of a magma would be high magnesian olivine.

This has been slightly rephrased to emphasize the fact that relatively easy ascent through a
thinner crust should reduce the extent of crustal processing, and therefore southern pSMI should
present a closer approximation of primitive mantle-derived magmas.

Line **126:** Increase in Cu with progressing magma differentiation is only possibly the case for Villarica.

Rephrased as ‘the general lack of Cu depletion’.

Line **127**: High pressure degassing of S and Cl is quite exotic process, and the modeling should involve
H₂O and CO₂ components in the magma. The authors could use SolEx program for more realistic
modelling (Witham, F., Blundy, J., Kohn, S.C., Lesne, P., Dixon, J., Churakov, S.V., Botcharnikov, R., 2012.
SolEx: A model for mixed COHSCI-volatile solubilities and exsolved gas compositions in basalt.
Computers & Geosciences 45, 87-97).

The SolEx model is based on experiments using low fluid/melt mass ratios and where fluid
compositions were calculated by mass balance based on run product glass compositions. Therefore Cl
partition coefficients are often poorly constrained, in particular for mafic melts where the fluid/melt
partition coefficients are relatively low (e.g. see argumentation in Zajacz et al. 2012, GCA). This then
propagates through SolEx to the effect that Witham et al. warn that estimates of the Cl in concentration
in modelled fluids may be accompanied by hundreds of percent relative error and so caution is
warranted when using SolEx to model Cl. We are therefore hesitant to make interpretations based on
this program, because the Cl concentration in the fluid will be very important for evaluating any
resulting perturbation of the Cl budget in the melt related to degassing (e.g., the SMI data). We have
added a note (lines 191-192) acknowledging more explicitly the lack of good experimental data to model
Cl degassing from similar magmas and pressures.

Nevertheless, we conducted additional MELTS modelling (Ghiorso and Gualda, 2015, Contrib
Min Pet) using a basaltic Villarrica pSMI melt composition to constrain H₂O and CO₂ systematics. Results
indicate that even with modest inferred pre-degassing CO₂ concentrations for arc basalts (~4500 ppm),
an aqueous fluid will saturate by the time the magma has ascended to mid-crustal levels (~700 MPa).
We note also that relatively water-rich primitive arc magmas (as may well be the case for Villarrica
especially) should also reach H₂O saturation by the mid crust when accounting for the increase in H₂O
concentration accompanying the expected extent of differentiation. Importantly, the saturation
pressure for a mixed CO₂-H₂O fluid will be higher than when considering CO₂ or H₂O separately. A much-
expanded discussion of this has been added to the text at lines 177-192.

Geochemically similar effects can also result from low-pressure mixing of primitive and evolved
magmas such as in Tolbachik volcano in Kamchatka (e.g. Kamenetsky et al., 2018; Chem Geology, Fig.
13).

We agree that mixing during the assembly of multiple magma batches in the shallow crust can
generate the observed variability in the SMI compositions from the SVZ. However, the key point that we
emphasize in the paper is that the variation in the compositions of the magma batches must be
generated at higher pressure before their arrival in the upper crust. Note that differentiation at upper
crustal depths cannot explain the strong increase in K₂O and other incompatible elements while the
concentration of other major elements (e.g. SiO₂) and the Mg# of mafic host minerals change only
moderately. The same problem holds for an explanation involving the mixing of an evolved melt
generated chiefly by crystal fractionation in an upper crustal magma reservoir with a more primitive
melt arriving from greater depth. This is because the felsic member participating in the mixing process
would have had to have much higher SiO₂ and Fe/Mg to attain the required high incompatible trace
element and K₂O concentrations as in the cool upper crust the principle mechanism of differentiation is
crystal fractionation without significant crustal assimilation.

Line **141**: Extensive assimilation is very energy consuming and implies extensive crystallization. There
must be some correlations of volatiles and trace elements against Fo-number of olivine.

A new Extended Data Figure (5) has been added which summarizes such correlations. Both fluid
mobile (K) and immobile (Nb) incompatible element concentrations in the SMI increase rapidly as host
529 Mg# decreases, and the rate of increase at Maipo and San Jose is even higher than at the other
volcanoes to the south (consistent with increasing crustal assimilation).

Dzierma et al. (2012) is more appropriate reference here than ref. 33.

Now cited instead, as suggested.

Line **157-158**: Ideally the authors should demonstrate sulfide inclusions in olivine in the northern
magmas. Else this is simply speculation.

See response to comment on Line 161-181 below.

Line **158-160**: Can it be a primary feature inherited from low-T mantle melting? How can low-T mantle
melting be ruled out as the control?

It is challenging to generate primary mantle melts with such low (≤ 60 ppm) Cu concentrations.
Our model indicates that this would require quite a low solubility of S^{2-} in the mantle melt (SCSS ≤ 1000
543 ppm) and quite a high amount of mantle sulfide (equivalent to ≥ 350 ppm S). Furthermore, this is for the
544 primary melt, and Cu should have been further enriched during magma differentiation in the lower crust
if no sulfide saturation took place. A primary melt with 40 ppm Cu could contain 60 ppm Cu after
differentiation to a mafic-intermediate SVZ composition, but the production of such a primary melt
would require very low S^{2-} solubilities in the silicate melt (SCSS = 700 ppm) and very high mantle sulfide
abundance (equivalent to 550 ppm S). The proposition of a low-T mantle melting explanation is unlikely
on this basis, and also in light of our newly added argumentation using Cu/Ag ratios, which show
agreement between the SMI compositions and our presented sulfide modelling.

Line **161-181**: I think there are many sources of uncertainty involved in this calculation and it is not clear
why this estimate is important. Again, the best argument for sulfide crystallization would be photo of
sulfide inclusion in primitive olivine.

We are not proposing that the SMI were trapped at lower crustal conditions but rather that the
olivine grew and entrapped the SMI following melt ascent. Therefore, one would not expect lower
crustal sulfide (especially if physically fractionated from the magma) to be present as an inclusion in the
studied olivines. This is also why the estimate is important: we argue that these SMI represent the
nearest approximation of magma compositions generated by differentiation at mid to lower crustal
depths, and therefore the SMI volatile and ore metal budgets (especially Cu and Au) as utilized in this
calculation offer an opportunity to constrain and quantify the mass and character (sulfide vs liquid) of
the lower crustal sulfide. These estimates are of general importance because of the difficulty of directly
sampling the lower continental arc crust, and of direct relevance when attempting to constrain the
capacity of sulfide-rich lithologies to augment the metal and S budgets of ore forming magmas.

Line **190-191**: Again, I am not convinced that the low Cl and S result from degassing during magma
fractionation, instead of from primary magma and/or from low pressure magma mixing.

Addressed in various other responses and revisions, emphasizing that while variably degassed
magmas do mix prior to eruption, the processes leading to relatively Cl-depleted batches must have
occurred at deeper crustal levels.

Line **192-193**: “hinders efficient ore mineral precipitation” Sulfide crystallization?

Yes, but as sulfides precipitated as mineralization within a hypothetical shallow hydrothermal
system. This sentence now specifies that a lack of S in the fluid specifically hinders precipitation of ore
metal sulfides in the associated hydrothermal system to avoid any confusion.

Line **202-207**: I partly disagree with this conclusion as explained above.

Addressed in responses and revisions.

METHODS:

- It should be explained somewhere that two different datasets are shown on figures: LA-ICP-MS data
for unexposed inclusions and 2) EPMA for exposed inclusions. It would be good to demonstrate the
consistency of these two data sets for some elements!

Figure 2 and 3 captions have been updated to state more clearly that Cl was measured via EPMA.
New Extended Data Fig. 3 shows the good agreement between the two methods, as does Fig. R1 below.

- Line **357-360**: there is detectable signal of S and K during LA-ICP-MS olivine analysis (ext data Fig.2). Is
this real?

For S, this is the ‘artificial’ signal mentioned in the Methods, whereas the K signal may be real as it
generally amounts to $\ll \sim 20$ ppm K_2O . Note that the artificial S signal does not affect the quantified SMI
compositions as it is removed by the host correction procedure during the quantification of the SMI
compositions (see Methods).

- All LA-ICP-MS analyses are done using GSD-1g glass as standard. Are olivine analyses stoichiometric?
Can it be demonstrated?

It can be demonstrated that LA-ICP-MS and EPMA analyses return comparable olivine
composition populations, which validates the LA-ICP-MS analyses.

Fig. R1. Comparison of FeO and MgO in olivine populations from Planchon (a) and Villarrica (b), as
measured by both LA-ICP-MS and EPMA.

**DATA:**

There are some problems with Pt - poor data for standards BHVO-2 and TDB-1. The other elements
important for this work - Au, Ag and Cu seem to be ~ Ok.

We attribute the lower than expected Pt concentrations measured in glasses made from these
powdered samples to minor Pt nugget formation during the fusing process. This is noted explicitly now
in the table, and does not affect any interpretations or conclusions of the study as these are based
largely on SMI rather than whole rock compositions.

The source of reference data for TDB-1 should be checked; most elements were not reported in the
cited source.

When we follow the provided URL and cited reference, the corresponding web page and the
available PDF both report all values present in our table.

Sc and Cu data in olivine should be corrected for interference with SiO on mass 45Sc and MgAr on mass
65Cu. If not corrected then it must be indicated or data excluded.

We did not correct for these interferences and justify that choice with the following reasoning.

In general, argide interference production rates were likely very low during our analyses. We
base this assertion on our specific tests for the contribution of argide interferences to Au and Pt signals
(see Methods). This suggests that MgAr interference on the Cu signal, too, is negligible, and application
of the Au and Pt argide production rates (~0.0001 %) to olivine measurements yields less than 0.1 ppm
Cu. As additional supporting argumentation, any significant interference-derived Cu that should be
apparent in Mg-rich phases like olivine, would drive the ratio of Cu in olivine/melt to artificially high
values, and should also be positively correlated with the Mg concentration of the host. Such trend is not
apparent in our dataset. Furthermore, prior measurements of both 63Cu and 65Cu in mafic-
intermediate glasses with 10x different Cu concentrations (~3 wt % MgO and ~40 ppm Cu; ~5-6 wt %
MgO and 3-6 ppm Cu) yielded an average absolute difference between Cu concentrations derived by
either isotope of 0.36 ± 0.30 ppm. These measurements were done with the same instrument tuned to
the same measurement conditions. Finally, the measured ratio of Cu in olivine/silicate melt (mostly
< 0.06) is similar to or even less than experimentally derived Cu partition coefficients, including those
from experiments with > 100 - 1000 's ppm Cu in each phase (i.e. in which the derived partition
coefficients will likely be unaffected by minor polyatomic interferences; e.g., Liu et al. 2014, GCA).

Regarding SiO interference on Sc, the most sensitive phase for interference on Sc will be that
with the lowest Sc, e.g. olivine, but even for olivine a generally expected contribution of artificial Sc
signal via SiO amounts to only ~0.5 ppm with an LA-ICP-MS tuned to similar oxide production rates as
ours (ThO/Th $< 0.4\%$; Jenner and O'Neill 2012, GGG). This would be only a minor contribution to the
typical Sc concentrations we measured in olivine (~5-10 ppm). Furthermore, like Cu, the measured Sc
olivine-melt D values are a good match for literature values (e.g., Mallmann and O'Neill, 2013, JPet).

Also, please note that we do not utilize the trace element compositions of olivine in this study
and the very small possible absolute errors on these two elements in the olivine would propagate to a
negligible relative error on the SMI compositions as the melt is much richer in these elements.

Much data on olivine host have elevated Ba (above DL), which correlates with Na, K, Sr, LREE and other
highly incompatible in olivine elements and indicates contamination by LILE-LREE rich glass.
All data must be checked before publication, and analyses affected by the contamination (Ba above
detection limit) should be clearly indicated and affected elements excluded.

The measured above-detection Ba concentrations are largely related to the very low detection
limits (<1 ppm). Figure R2 below shows an example LA-ICP-MS analysis of an especially ‘high-Ba’ olivine
with >1 ppm Ba. The incompatible element signals are free of localized peaks that would suggest
transient signal contamination. Importantly, unlike Ba and Sr, several other incompatible elements
shown in Fig. R2 (e.g., Ti, Mo, Yb, Y, P) are not enriched relative to ‘typical’ olivine from the sample. For
additional context, pairing the olivine and SMI Ba concentrations yields average olivine-melt partition
coefficients of 0.0014 ± 0.0014 , with a median of 0.0010 and maximum of 0.0070 through the full
dataset. Thus even those few olivine analyses with 1-3 ppm Ba (13 out of 141 total analyses) are in
agreement with the expected highly incompatible nature of Ba. The precise cause of these rather
strange Ba concentrations remains unclear, including whether some subtle local circumstances led to Ba
contamination during these few analyses, but ultimately this does not affect any of the conclusions or
interpretations that we present.

Fig. R2. Example LA-ICP-MS analysis for a ‘high-Ba’ (1.67 ppm) olivine from San Jose.

After reinspection of our data, one clinopyroxene analysis (row 33, Maipo, sample 17-CG-012e)
has been noted as having “possible contamination in the host signal by minor incompatible element-rich
phase” because of its rather high concentration of some nominally highly incompatible elements (e.g. Rb
and Pb).

REVIEWER COMMENTS

Reviewer #1 (Remarks to the Author):

Dear Editor,

I read the revised manuscript plus the responses to all reviewer comments and found all comments satisfactorily addressed. I thus recommend the manuscript for publication in its present state.

Sincerely

Reviewer #2 (Remarks to the Author):

My main concerns related to the previous submission were related to the depth at which volatile saturation occurred, and the unrealistically elevated water contents required to achieve saturation in the lower crust. This aspect of the study has now been removed and a different and valid explanation has been provided. I concur with this change.

Reviewer #3 (Remarks to the Author):

Comments: "Sulfur and chlorine budgets control the ore fertility of arc magmas"

After reading the revised manuscript, I am still not convinced that Cl and S are lost in the lower crust in the southern SVZ magmas. They can also reflect low initial concentrations from the mantle. Overall, the authors seem to have stuck to their (very speculative) model and do not consider alternative scenarios.

Publication of this paper would be "useful speculation", but I do not agree with the interpretation (and overall style of addressing the reviewers' comments) and believe there is large unexplored potential of this dataset.

Title: As stated very strong and doesn't allow for other possibilities. I would suggest rephrasing as "Sulfur and chlorine budgets: an important control on the ore fertility of arc magmas".

Lines 18-20, 174-177, 302-305: "Magmas in the north are, however, S and Cl rich, unlike those to the south..." This statement is not correct as written, since Wehrmann et al. (2014) show that magmas further south than the study area (41-42°S) also have elevated S (between 2000-2500 ppm) and Cl (up to 1400 ppm) with similar concentrations to those in the north (north of 34°S). In addition, Los Hornitos (35.5°S) in the authors' "southern group" also has elevated S (up to just over 3000 ppm) and Cl (up to 1300 ppm). Therefore, there is not a simple decrease in S and Cl from north of 34°S and south of 34°S. Crustal conditions and compositions of the melts (e.g. how primitive they are) between Villarica (39-40°S) and Cabo de Vaca and Apagado (41-42°S) are very similar, so what causes these variations in S and Cl? As pointed out by the authors, there is a fracture zone subducting beneath Villarica. Thus, there is a higher fluid flux from lithospheric mantle serpentinized by seawater of the down-going slab than beneath other volcanoes south of 34°S. What influence will fracture zone subduction have on not just water but also S and Cl contents? In general variations in slab input are ignored, but are certainly also likely to play a role in S and Cl budgets of arc volcanics.

Line 45. How do the melt inclusions preserve a "time-resolved record"? Since this is not important for the paper, I suggest deleting this text.

Line 78. "Gold and Pt concentrations show similar patterns to Cu" No. Gold shows similar concentrations at the southern volcanoes of Hornitos and Antuco to the northern volcanoes,

whereas Cu is higher for all southern compared to northern group volcanoes. With one exception, the range in Pt is similar between northern and southern group volcanoes, actually at Antuco the Pt concentrations are lower. Considering that there are only two analyses in the northern group, this clearly cannot be considered to be a representative data set and thus statements about the similarity of Pt to Cu should be avoided!

Line 79. Well either the patterns are similar or they aren't, but how can they be "partially obscured by higher analytical uncertainties"? I don't follow this.

Lines 84-86. As noted above, not seen in the Wehrmann et al. (2014) dataset. Also, as noted previously, why not include the Wehrmann et al. data on the plots in figure 2! This data is also from olivine melt inclusions and certainly together with the authors' data would give a more representative view of variations in these volatiles along the arc!

Lines 86-89: There seems to be contradiction between Fig 1 caption "Sulfide saturation-related scavenging of Cu, Au, and Pt occurs during crustal thickening, despite an accompanying shift to slightly more oxidizing conditions" and lines 86-89 "The absence of magmatic sulfides in all studied samples except those from San Jose and Maipo (within which they are present in only trace abundance; Extended Data Table 2) suggests the observed variation in S and chalcophile ore metal concentrations is not induced by this phase during upper crustal magma storage." Where is the evidence then for sulfide fractionation in the lower crust? The discussion should be based on the available data.

Pages 6-8 and rebuttal to reviewer's comments: The authors continue not to show and do not use olivine Fo-number to evaluate extent of melt fractionation. I cannot agree with this approach and suspect the authors may be hiding important information or they would show this. K₂O is not good proxy of magma fractionation, because it also varies with slab input and assimilation, and unless strong negative correlation with magma or olivine Mg# exists, K₂O should not be used for this purpose!

Line 118-119. Are these correlations statistically valid? I doubt it. Therefore, the authors should either include "slightly" or "crude" positive correlations.

Line 166: An aqueous fluid at 1-2 GPa can't be in equilibrium with the melts. The question was already raised previously!

Line 174: The best argument for early Cl degassing would be a figure showing decreasing Cl concentration with decreasing Fo of olivine. Otherwise, this is purely speculation or perhaps evidence for assimilation of low Cl crustal material.

Lines 201-204: Villarica is clearly an end member, but because it is underlain by a fracture zone how do the authors distinguish between different S and Cl inputs from the slab and "degassing prior to arrival in the upper crust".

Line 208-210: Why couldn't Llama have a different initial K/Cl than Villarrica?

Lines 216-218: "unlikely consequence of slab-derived element flux" Why?

Lines 235-237: There is also a crude positive correlation between K₂O and S. What implication does this have?

Lines 249-263 and Fig. 5: A very nice addition to the manuscript; however, I again question to what extent slab input plays a role. For example, slab or sediment melts could have very different Cu/Ag ratios. What role do they play? For example, slab melts are likely to have low Cu/Ag ratios and could serve as a mixing end member in the north, suggesting an alternative to sulfide fractionation.

Fig. 5. Analysis of Cu/Ag is valuable addition. However, it is still unclear if the variations reflect sulfide related fractionation in the crust or in the mantle. Cu/Ag should be plotted versus Fo

number of olivine and also vs. Latitude in Fig. 1.

Fig. 5. Mo behaves as strongly compatible element in this model. Does it agree with its good correlation with Ce in MORB magmas (Kelley et al 2013)?

Finally, concerning olivine composition: The authors must demonstrate how well the olivine standard was analysed by their approach or accept reviewer comments and exclude questionable elements (at least Sc, Cu). Their arguments based on "should be correct" and "should not be affected" are not acceptable! The data is not used in the manuscript, but incorrect data should not be published! Strong correlation of Ba with other incompatible elements in olivine strongly suggest contamination by melt glass phase during analysis. I am very surprised about the authors' insistence of the opposite. Isn't it more constructive to delete contaminated analyses?

NCOMMS-20-19898-T (by Grondahl and Zajacz)

RESPONSE TO REVIEWER COMMENTS (Round 2)

Please see our responses in blue following each comment.

Reviewer #1 (Remarks to the Author):

Dear Editor,

I read the revised manuscript plus the responses to all reviewer comments and found all comments satisfactorily addressed. I thus recommend the manuscript for publication in its present state.

Sincerely

Reviewer #2 (Remarks to the Author):

My main concerns related to the previous submission were related to th depth at which volatile saturation occurred, and the unrealistically elevated water contents required to achieve saturation in the lower crust. This aspect of the study ahs now been removed and a different and valid explanation has been provided. I concur with this change.

Reviewer #3 (Remarks to the Author):

Comments: "Sulfur and chlorine budgets control the ore fertility of arc magmas"

After reading the revised manuscript, I am still not convinced that Cl and S are lost in the lower crust in the southern SVZ magmas. They can also reflect low initial concentrations from the mantle. Overall, the authors seem to have stuck to their (very speculative) model and do not consider alternative scenarios.

Publication of this paper would be "useful speculation", but I do not agree with the interpretation (and overall style of addressing the reviewers' comments) and believe there is large unexplored potential of this dataset.

Title: As stated very strong and doesn't allow for other possibilities. I would suggest rephrasing as "Sulfur and chlorine budgets: an important control on the ore fertility of arc magmas".

The current version of the title is to the point and expresses the main message of the manuscript.
Although we understand the reviewer's concern, we think it is almost self explanatory that though we
emphasize the important role S and Cl plays in magmatic ore fertility, other factors may play a role as well.
Therefore, we would prefer to keep the present version of the title.

Lines 18-20, 174-177, 302-305: "Magmas in the north are, however, S and Cl rich, unlike those to the
south..." This statement is not correct as written, since Wehrmann et al. (2014) show that magmas
further south than the study area (41-42°S) also have elevated S (between 2000-2500 ppm) and Cl (up to
1400 ppm) with similar concentrations to those in the north (north of 34°S). In addition, Los Hornitos
(35.5°S) in the authors' "southern group" also has elevated S (up to just over 3000 ppm) and Cl (up to
1300 ppm). Therefore, there is not a simple decrease in S and Cl from north of 34°S and south of 34°S.
Crustal conditions and compositions of the melts (e.g. how primitive they are) between Villarica (39-
40°S) and Cabo de Vaca and Apagado (41-42°S) are very similar, so what causes these variations in S and
Cl? As pointed out by the authors, there is a fracture zone subducting beneath Villarica. Thus, there is a
higher fluid flux from lithospheric mantle
serpentinized by seawater of the down-going slab than beneath other volcanoes south of 34°S. What
influence will fracture zone subduction have on not just water but also S and Cl contents? In general
variations in slab input are ignored, but are certainly also likely to play a role in S and Cl budgets of arc
volcanics.

Our statement that magmas to the south are less S and Cl rich than Maipo and San Jose is correct
because it is describing the samples presented in this study. Regarding the highest S and Cl concentrations
from Los Hornitos, the reviewer appears to have confused our *pSMI* (Fig. 2) and *fSMI* (Fig. 3) datasets. Our
statement of a southward trend is based on *pSMI* as presented in Fig. 2, and there it can be seen that Los
Hornitos *pSMI* fit the trend well.

Regarding Cabo de Vaca (assumed to be 'Cabeza de Vaca' in the Wehrmann et al. dataset) and Apagado,
we decline to comment on them in this paper because they are located far outside of our study area
(south of Villarrica by ~200 and ~275 km, respectively). Furthermore, as also noted by Wehrmann et al.,
these two volcanoes are small monogenetic cones analogous to the similarly volatile-rich Los Hornitos
SMI rather than to Villarrica. The monogenetic character of Los Hornitos is mentioned in the Fig. 3 caption,
where the reader is referred to the Supplementary Discussion for additional discussion, and in the
Supplementary Fig. 1 caption. Note that S and Cl from the three larger (and closer) 'polygenetic'
stratovolcanoes south of Villarrica in the Wehrmann et al. dataset are much more similar to Villarrica than
to the distant monogenetic cones. The polygenetic – monogenetic distinction is important in the
framework of our study because monogenetic volcanoes in the Andes and elsewhere generally deliver
rather primitive mantle melts to the surface without much differentiation. Magma differentiation in the
crust is a common process among the other volcanoes and forms a key part of our interpretations.
Therefore, our statement generalizing the southward trend remains valid in this context as well.

Slab input variations are not ignored, and were discussed at lines 241-254 as one of the processes that
control the initial volatile budgets of the studied volcanoes. It is indeed possible that dehydrating a
fracture zone-bearing slab could have an impact on S and Cl budgets of the resultant mantle melts.
However, judging by the coherency of Villarrica *pSMI* with the demonstrated trend in Fig. 2, this does not
appear to have significantly affected the initial S and Cl budgets.

Line 45. How do the melt inclusions preserve a “time-resolved record”? Since this is not important for
the paper, I suggest deleting this text.

Removed as suggested.

Line 78. “Gold and Pt concentrations show similar patterns to Cu” No. Gold shows similar concentrations
at the southern volcanoes of Hornitos and Antuco to the northern volcanoes, whereas Cu is higher for all
southern compared to northern group volcanoes. With one exception, the range in Pt is similar between
northern and southern group volcanoes, actually at Antuco the Pt concentrations are lower. Considering
that there are only two analyses in the northern group, this clearly cannot be considered to be a
representative data set and thus statements about the similarity of Pt to Cu should be avoided!

Please see our response to the following comment regarding Line 79.

Line 79. Well either the patterns are similar or they aren’t, but how can they be “partially obscured by
higher analytical uncertainties”? I don’t follow this.

These lines concerning Au and Pt have been rephrased according to the reviewer’s concerns
expressed in this and their comment regarding Line 78. We now state that because of their chalcophile
nature, Au and Pt would be expected to display similar latitudinal trends to Cu, but no longer imply that
this is clearly the case (lines 104-107). We have also expanded the cautionary note stating that
interpreting Au and Pt is challenging for the reason that relatively large analytical uncertainties
accompany their very low concentrations.

Lines 84-86. As noted above, not seen in the Wehrmann et al. (2014) dataset. Also, as noted previously,
why not include the Wehrmann et al. data on the plots in figure 2! This data is also from olivine melt
inclusions and certainly together with the authors’ data would give a more representative view of
variations in these volatiles along the arc!

As discussed in the Fig. 2 caption, S and Cl concentrations from the ‘least-degassed’ Wehrmann
SMI are generally quite similar to SMI from our study, except for their lower S concentrations in SMI from
Maipo (likely due to post-entrapment migration of S from the melt phase into the bubble within the
closed-system SMI, yielding erroneously low S concentrations in EPMA measurements). However, not all
of the relevant volcanoes in the Wehrmann dataset yielded SMI which pass our pSMI filtering criteria, and
so not all volcanoes would have Wehrmann SMI plotted along side our data. Adding this similar but
incomplete dataset to our figure would merely show that S and Cl has previously been measured in SMI
from the SVZ, and this is a point we have already made in the figure caption. There would be no extra
insight gained regarding volatile concentration variations along the arc (please also see our response to
the lines 18-20 comment above). Therefore, our preference is to keep the version of the figure that shows
only our own data, but upon request we can replace it with the version shown here (Fig. R1).

Fig. R1. Example of Fig. 2 with the data of Wehrmann et al. (2014) added as black x symbols offset to the
 right, following pSMI filtering of their ‘least degassed’ SMI data set.

Lines 86-89: There seems to be contradiction between Fig 1 caption “Sulfide saturation-related
 scavenging of Cu, Au, and Pt occurs during crustal thickening, despite an accompanying shift to slightly
 more oxidizing conditions” and lines 86-89 “The absence of magmatic sulfides in all studied samples
 except those from San Jose and Maipo (within which they are present in only trace abundance;
 Extended Data Table 2) suggests the observed variation in S and chalcophile ore metal concentrations is
 not induced by this phase during upper crustal magma storage.” Where is the evidence then for sulfide
 fractionation in the lower crust? The discussion should be based on the available data.

We see no contradiction between these two points. The scarcity of sulfide saturation in the upper
 crust does not mean that this phase could not have been present in the lower crust. The shift to more
 oxidizing conditions is not to a degree that makes a smaller coexisting proportion of reduced sulfur
 implausible. In addition, increasing pressure and decreasing temperature shift the sulfide to sulfate
 transition to higher fO_2 and this may further contribute to sulfide saturation in the north at lower crustal
 depths. Furthermore, please note that the sulfur concentration in the silicate melt at sulfide saturation
 increases with decreasing pressure and arc magmas most typically increase their oxidation state with
 progressing differentiation. Therefore, the absence of sulfide saturation upon arrival in the upper crust is
 not in conflict with sulfide saturation at lower crustal depths.

As explained in the previous round of responses to the reviewer’s comment on then-lines 158-160, and
 as shown in Figure 4, it is quite difficult for mantle melting to generate a mafic arc magma with Cu
 concentrations as low as those measured in the northern SMI. Therefore, Cu-poor magmas must have lost
 Cu at some point after initial magma genesis. The assumption of cryptic sulfide saturation at depth to
 explain variation in whole-rock Cu concentrations is common in the literature, as explained in the
 introduction (e.g. Lee et al., 2012, Science, Chiaradia, 2014, NGeo; Jenner, 2017, NGeo). The SMI data
 presented in this manuscript, and the model calculations based on it, provide a more direct line of
 evidence for lower crustal sulfide saturation as we are looking at a record of volatile element and

chalcophile metal concentrations that is not affected by syn-eruptive degassing (lines 265-285, and lines
728-739 in the Methods).

Pages 6-8 and rebuttal to reviewer's comments: The authors continue not to show and do not use
olivine Fo-number to evaluate extent of melt fractionation. I cannot agree with this approach and
suspect the authors may be hiding important information or they would show this. K₂O is not good
proxy of magma fractionation, because it also varies with slab input and assimilation, and unless strong
negative correlation with magma or olivine Mg# exists, K₂O should not be used for this purpose!

In contrast to the reviewer's assertion that we do not show olivine Mg# and are 'hiding important
information', host mineral Mg# was shown in the previously added Supplementary Fig. 5, plotted against
fluid mobile (K₂O) and immobile (Nb) elements. This was stated in our response to their previous comment
on then-line 141. There is indeed an overall negative correlation between both of these elements and
host-Mg#, and this is especially strong at San Jose, Maipo, and Villarrica. Collectively, two clear groups are
discernable which correlate to our 'northern' and 'southern' groupings. We now explicitly direct the
reader's attention to this plot to avoid further confusion (lines 222-225).

As we explained in responses to similar comments from this reviewer during the previous round of
revisions, the sensitivity of K₂O to assimilation is a *desirable* trait which yields a more complete picture of
the extent of differentiation than Mg#, because in general this process is driven by both assimilation *and*
crystal fractionation. It is difficult to dominantly attribute the K₂O variation *within* an individual volcano
to slab input variability because that implies that large variations in the composition of the slab-derived
component occur over geologically short timescales at a single location (now explained explicitly at line
247-254; see also our response to comments on lines 216-218). However, we do agree that the influence
of the slab fluid composition could be reflected in differences *between* different volcanoes and their
primitive magmas, and have now explicitly mentioned K₂O in statements added to address this during
previous revisions (e.g., lines 241-246). It is worthy to note though that K₂O concentrations in the melt
inclusions seem to converge toward a similar value with increasing host olivine Mg# as shown on
Supplementary Fig. 5, and both fluid-mobile (K₂O) and -immobile (Nb) elements exhibit the same
distribution as a function of Mg#. This indicates that the variation in incompatible element concentrations
is primarily induced by the incorporation of a crustal melt rather than a slab-derived fluid. We now
emphasize this in the manuscript at lines 222-225 and 252-254.

Furthermore, the physical processes of assimilation and crystal fractionation are linked because
maintaining heat balance during assimilation requires crystallization, and therefore assimilation also
influences olivine Mg# in addition to magma composition. Note that in Supplementary Fig. 5, host mineral
182 Mg# is only well correlated with incompatible element concentrations in the SMI at certain volcanoes (e.g.
Villarrica, Antuco and Maipo), whereas at other volcanoes, host olivines spanning a narrow range of Mg#
contain SMI with a wide range of incompatible trace element concentrations. Significant ranges in both
fluid-mobile and -immobile elements are overall not correlated with host Mg#, except at Villarrica (and
perhaps Antuco). This latter observation implies that the Mg# of the host mineral was either buffered
during crystallization, or that primary Mg# variations were later homogenized and/or overprinted by
diffusive Fe-Mg exchange with the surrounding silicate melt. In either case, the host Mg# is not the best
proxy for differentiation, as elaborated upon below.

First, please consider that the Mg# of the minerals crystallizing and trapping SMI in the upper crust are
strongly related to the Mg# of the silicate melt (i.e. olivine Mg# is fixed by the equilibrium olivine-melt Fe-

192 Mg exchange coefficient, or k_D , of ~ 0.3 ; Ulmer, 1989, CMP). Accordingly, the use of mineral Mg# as a
193 proxy for differentiation is underlain by the assumption that the melt Mg# will always progress to lower
values as differentiation proceeds due to the prevailing fractionation of mafic minerals with high Mg/Fe
ratios. However, one must consider that melts differentiating in the lower crust will be in contact with
mafic cumulate crystal mushes and country rocks, to the effect that chemical buffering of melt Mg# may
occur (e.g., Reiners, 1998, JPet; Dungan and Davidson, 2004, Geology; Getsinger et al., 2009, JPet;
Melekhova et al., 2017, CMP; Cashman et al., 2017, Science; Cooper et al., 2019, CMP). These magma
batches may also be periodically recharged by more primitive melts with the important effect that after a
few recharge events, bulk-compatible elements will exhibit less variation than bulk-incompatible
elements. Therefore, compatible elements (e.g. Mg and Fe^{2+}) are less sensitive to differentiation during
these recharge-punctuated differentiation cycles (Chiaradia et al. 2011, JPet; Lee et al., 2014, GCA;
Portnyagin et al., 2015, JVGR).

These processes can effectively decouple major elements, including the Mg# of both the melt and
subsequent upper crustal SMI host minerals, from the differentiation trends preserved in incompatible
element budgets. This bolsters the appeal of using incompatible element variations as a measure of
differentiation within individual volcanoes, as has been done by many researchers in this field (e.g.,
Dungan and Davidson, 2004, Geology; Zajacz and Halter, 2009, EPSL; Audetat et al., 2011, JPet; Bouvet
de Maisonneuve et al., 2012, JVGR; Rottier et al., 2020, JVGR). It also supports the approach of basing
interpretations of S and Cl behaviour on the behaviour of incompatible elements such as K_2O rather than
211 Mg/Fe ratios. Deviations from the expected positive correlation between K_2O and volatile elements can
then be attributed to processes which could strongly fractionate these incompatible elements from one
another (e.g. magma degassing and sulfide/sulfate saturation).

Finally, a K_2O -based differentiation proxy is also advantageous because it is far more difficult to modify
the K_2O concentration in an SMI via diffusive re-equilibration with the ambient silicate melt outside the
host mineral, compared to the ease of resetting the Fe/Mg ratio in olivine and other mafic minerals. Fe-
217 Mg interdiffusion in olivine is very fast (Costa and Dungan, 2005, Geology; Gordeychik et al., 2018,
NSciRep). We modelled the time necessary to re-equilibrate the Mg# of olivine at conditions relevant for
upper crustal magma storage in volcanic arcs ($T=1100$ °C, $P=200$ MPa, $fO_2=NNO+1$; via DIPRA software:
Girona and Costa, 2013, GGG). Even a relatively large (~ 1 mm) olivine crystal with initial Mg# of 90 would
completely re-equilibrate to an Mg# corresponding to equilibrium with the surrounding silicate melt (e.g.,
222 Mg#=80) in ~ 80 years, which is a short timescale even for magma storage in the upper crust.

For all the above reasons, we prefer to continue employing K_2O rather than Fo as our differentiation proxy.

Line 118-119. Are these correlations statistically valid? I doubt it. Therefore, the authors should either
include "slightly" or "crude" positive correlations.

These manuscript lines concern Villarrica. The correlations referred to here for each of Cu, S,
and Cl with K_2O , which are statistically significant linear correlations at a 95% confidence level ($p<0.05$).
We have therefore left the wording unchanged.

Line 166: An aqueous fluid at 1-2 GPa can't be in equilibrium with the melts. The question was already
raised previously!

The questioned line in the manuscript demonstrates that although high-pressure degassing and
concurrent Cl partitioning between fluid and melt is currently an area needing greater experimental
constraint (we explicitly note this at lines 219-220), the best available experimental evidence suggests
that degassing at elevated pressures is a plausible mechanism to reduce Cl concentrations in hydrous arc
magmas.

Furthermore, nowhere do we invoke Cl scavenging by fluid saturation at near-2 GPa pressure and in fact
we argue against degassing at >1 GPa (lines 275-276). We addressed mid- to lower-crustal degassing in
the previous round of revisions in the following ways: 1) The plausibility of fluid saturation at ≤ 1 GPa
(reasonable lower-crustal storage conditions for Villarrica magma) has received much additional
supporting argumentation in the previously revised manuscript (lines 205-220); 2) the language was
revised to invoke simply 'greater than upper-crustal' pressure; 3) specific modelling support was added to
demonstrate the possibility of degassing at ~ 0.7 GPa, and; 4) the reviewer's detailed concerns were
addressed in responses during the initial round of reviews. The reviewer's comment here indicates that
she or he may have overlooked this, and therefore we refer to our previous revisions and responses, which
were found satisfactory by the other two reviewers who had similar concerns initially.

Line 174: The best argument for early Cl degassing would be a figure showing decreasing Cl concentration
with decreasing Fo of olivine. Otherwise, this is purely speculation or perhaps evidence for assimilation of
low Cl crustal material.

As explained at length in response to the reviewer's comments regarding pages 6-8, and
responses to previous rounds of comments, correlations with olivine Fo are not necessarily an ideal
approach and we prefer K_2O for the reasons stated. Nevertheless, at Villarrica, Cl and S are positively
correlated with Fo ($p < 0.05$; Fig. R2).

Figure R2. Supplementary Fig. 5, here shown also with S and Cl data added. A weak correlation exists
 between Cl and host mineral Mg# in Villarrica, but the same cannot be said about Planchon, Antuco, and
 Llaima. However, the olivine Mg# spans a narrower range at these latter volcanoes, and with the possible
 exception of Antuco, neither K₂O nor Nb show systematic relationships with Mg# either. In our
 interpretation, this is most likely due to diffusive re-equilibration of the Fe/Mg ratio of the olivine, or
 buffering of the melt Mg#, as discussed above.

Lines 201-204: Villarrica is clearly an end member, but because it is underlain by a fracture zone how do
 the authors distinguish between different S and Cl inputs from the slab and “degassing prior to arrival in
 the upper crust”.

We argue for degassing prior to arrival in the upper crust on the basis of the fSMI dataset, which
 shows S and Cl behaviour *during differentiation*. If the existence of the fracture zone resulted in special S
 and Cl budgets at Villarrica, this should already be apparent in the pSMI data because the influence of slab
 inputs is established prior to magmatic differentiation. However, as explained in response to comments
 on lines 18-20 and shown in Figure 2, Villarrica pSMI fall in line with the broad trend through the study
 area. Please also see our response to the comment on lines 216-218.

Line 208-210: Why couldn't Llaima have a different initial K/Cl than Villarrica?

This is possible, and we have added text (lines 238-240) clarifying that interpreting a deep-
degassing history at Llaima in this manner assumes that Llaima and Villarrica have similar initial K/Cl.

Lines 216-218: “unlikely consequence of slab-derived element flux” Why?

We have added text (lines 247-252) clarifying that to attribute the decoupling of K and Cl within
samples from an individual volcano to the nature of the slab derived component would require strongly
differing K/Cl ratios in this component over short timescales, and the preservation of these differences
from the point of release from the slab until final entrapment in the SMI in the upper crust.

Lines 235-237: There is also a crude positive correlation between K₂O and S. What implication does this
have?

The weak correlation between K₂O and S in the northern fSMI is *negative* (see Figure 3), and the
magnitude of the decrease in S correlates to our estimated S loss via sulfide fractionation as explained in
lines 280-283, and *Methods* lines 728-739.

Lines 249-263 and Fig. 5: A very nice addition to the manuscript; however, I again question to what
extent slab input plays a role. For example, slab or sediment melts could have very different Cu/Ag
ratios. What role do they play? For example, slab melts are likely to have low Cu/Ag ratios and could
serve as a mixing end member in the north, suggesting an alternative to sulfide fractionation.

Simply from a mass balance perspective, it is virtually impossible for plausible amounts of slab-
derived components to cause this trend, regardless of the fluid- or melt-like nature of the component and
the relative influences of the slab and sediments. For example, assuming a strong yet possible estimate
of slab-derived component introduction to the mantle wedge on the order of ~3 % relative, combined
with ~6 % mantle melting in the north, could yield a hybrid melt with up to a 1:2 ratio of slab component
: mantle melt. Even with a Cu/Ag ratio of zero in the slab component (which is difficult to imagine), Cu/Ag
in the hybrid melt would be decreased at most to ~1000 from the purely mantle melt value of ~1500,
which is still far higher than measured low end of ~500. Additionally, the ambient mantle sulfides would
buffer Cu (and to a lesser extent Ag) concentrations in melts ascending through the mantle. Please also
see our responses above with regards to slab input versus magma differentiation controls on the observed
chemical variations.

Fig. 5. Analysis of Cu/Ag is valuable addition. However, it is still unclear if the variations reflect sulfide
related fractionation in the crust or in the mantle. Cu/Ag should be plotted versus Fo number of olivine
and also vs. Latitude in Fig. 1.

As a generalization, the mantle wedge is thought not to be a site of magma fractionation, but
rather generation, and so the dominant mantle-sulfide influence on Cu/Ag would occur during mantle
melting. However, it would take unrealistically low degrees of melting (<1%, based on modelling
presented in Figure 4) to generate the lower range of observed ratios in the northern SMI. In contrast, we
show in Figure 5 that the Cu/Ag ratios are consistent with modelled extents of mantle melting,
differentiation, and sulfide fractionation throughout the dataset, in good agreement with our broad
conclusions.

We prefer not to add Cu/Ag to Fig. 1 as this is already a 6-panel figure, and the interested reader can see
that Cu/Ag tends to decrease northward overall even before seeing Figure 5 (as Cu decreases and Ag is

constant). We also elect not to plot Cu/Ag against olivine Fo (nor K_2O), because we have previously
established the behaviour of Cu during differentiation in Figure 3 and the relevant text, and it is apparent
in Figure 5 that Cu behaviour controls the inter-volcano variation of Cu/Ag when comparing Cu, the
latitudinally similar Ag, and Cu/Ag to the mantle-melt values (various bars and fields on Figure 5). Much
additional discussion of these general points is included in the caption of Figure 5.

Fig. 5. Mo behaves as strongly compatible element in this model. Does it agree with its good correlation
with Ce in MORB magmas (Kelley et al 2013)?

We do not quite understand this comment, as this figure and model do not concern Mo, nor is
there any implied reason to think Mo would behave strongly compatibly as opposed to the expected bulk
incompatible behaviour. Molybdenum and Ce are correlated in our dataset, although less tightly than the
MORB data presented by Kelley et al. (2013). This is likely due to the contamination of SVZ arc magmas
by crustal material of unknown and variable Mo/Ce.

Finally, concerning olivine composition: The authors must demonstrate how well the olivine standard was
analysed by their approach or accept reviewer comments and exclude questionable elements (at least Sc,
Cu). Their arguments based on “should be correct” and “should not be affected” are not acceptable! The
data is not used in the manuscript, but incorrect data should not be published! Strong correlation of Ba
with other incompatible elements in olivine strongly suggest contamination by melt glass phase during
analysis. I am very surprised about the authors’ insistence of the opposite. Isn’t it more constructive to
delete contaminated analyses?

We have now conducted additional LA-ICP-MS tests to assess the significance of possible mass
interferences on Cu and Sc concentrations in olivine. The results confirm the multiple lines of evidence
presented in our responses from the previous round of reviews. For Cu, we measured both ^{63}Cu and ^{65}Cu
in olivine. The Cu concentration derived from ^{63}Cu is not subject to any plausible mass interferences, which
would arise from either $^{47}Ti^{16}O$ or $^{23}Na^{40}Ar$. There is very little Ti in olivine (≤ 0.01 wt% TiO_2), and ^{47}Ti is only
$< 8\%$ of the natural Ti abundance. Likewise there is very little Na in olivine (≤ 0.01 wt% Na_2O), and tests
using NIST-617 show that even at very high Na_2O concentrations (14 wt%, $\sim 3-4x$ greater than even the
SMI) the production of $^{23}Na^{40}Ar$ contributes < 2 ppm to measured Cu concentrations. In olivine, the Cu
concentrations determined via ^{65}Cu are only 0.38 ± 0.07 ppm ($n=11$) greater than those using ^{63}Cu , for
total Cu concentrations of 5.3 ± 0.4 ppm vs. 4.9 ± 0.4 ppm, respectively. This is indistinguishable from our
estimate of 0.36 ± 0.30 ppm from our response to the previous round’s comment regarding this matter.
We have added a note to the Supplementary Table acknowledging this very small contribution.

We tested the importance of $^{29}Si^{16}O$ interferences on ^{45}Sc in olivine by comparing the concentration of Sc
in olivine measured when the LA-ICP-MS is tuned to routine nominal oxide production rates to Sc
concentrations measured at far higher oxide production rates than were acceptable in this study. The Sc
concentration at normal conditions (10.9 ± 0.4 ppm; $n=11$; $ThO/Th = 0.1-0.2\%$) is identical to that from
oxide production rates 5-10 times higher (10.6 ± 0.4 ppm; $n=12$; $ThO/Th = 0.9-1\%$). We therefore conclude
that the Sc concentrations we measured in olivine are not affected by this interference.

Regarding Ba, we did not insist that some minor contamination of the olivine signals could not have
occurred as the reviewer claims, and specifically acknowledged that possibility in our response to the
relevant comment. Additionally, we disagree that deleting these analyses is somehow more constructive,
because that would remove these otherwise fine analyses from the dataset. For example, to generate the

observed elevated concentrations of highly-incompatible elements in a generic high-Ba olivine (1.6 ppm
Ba, 0.15 ppm Rb, 0.6 ppm Zr) by mixing a typical olivine (0.4 ppm Ba, 0.03 ppm Rb, 0.1 ppm Zr) with an
SMI-like melt (480 ppm Ba, 50 ppm Rb, 200 ppm Zr) would require only ~0.25% addition of melt to the
analyzed olivine mass. This has very little effect on moderately olivine-incompatible elements like Sc, V,
and Cu (<10 % relative), and compatible elements would be diluted in the high-Ba olivine by less than the
0.25 relative %. We reiterate that the very slightly elevated incompatible element concentrations in this
small subset of olivine analyses does not impact any other data or conclusions (including, importantly,
SMI compositions).

Nevertheless, we agree that the readers should be notified of these potentially contaminated analyses
and we have flagged all the 'high-Ba' (>1 ppm Ba) olivine in the Supplementary Table as having possible
minor signal contamination by an incompatible element-rich phase, and note the absence of clear
evidence for this in the LA-ICP-MS signals (as discussed in our replies to the previous round of comments).

REVIEWER COMMENTS

Reviewer #3 (Remarks to the Author):

Comments on "Sulfur and chlorine budgets control the ore fertility of arc magmas" by Grondahl and Zajacz

The manuscript represents an improvement over the previous version; nevertheless, many issues have not been adequately addressed. The degassing of Cl-S in the south is unclear and not necessary to generate low Cl-S magmas. The discussion is extremely speculative. Also the criteria for "primitive" magma in this work is totally arbitrary, simply a mess, in this manuscript.

The major conclusion of the manuscript is that "the northern SVZ volcanoes generate world class porphyry Cu deposits because the thick crust facilitates the generation of prolonged volatile-rich magmatism where crucial volatile elements such as S and Cl are not significantly depleted in the magma prior to emplacement at upper crustal depths." It remains a contradiction in logic to me, why magmas would not lose S and Cl when traversing thicker crust in the north than in the south. The authors' provide two explanations for the behaviour of volatiles and ore minerals. They argue that increased degassing takes place at lower to mid crustal depths beneath the southern SVZ volcanoes, based on the "fluid/silicate melt partition coefficient of Cl ...strongly increasing with increasing pressure". Why would volcanoes in the south degas more than volcanoes in the north, particularly if the crust is thicker in the north. The second reason given in the manuscript for the decrease of S and Cl is the assimilation of plutonic roots. It is argued that the plutonic rocks have degassed their S and Cl, but not Cu and thus assimilation of the plutonic roots in the south will cause S and Cl to go down and Cu to increase. It is not clear to me, why older plutonic rocks will be preferably assimilated beneath the thinner southern crust than thicker northern crust. It seems more logical that these processes are more likely (or at least as likely) to occur under the thicker northern crust than beneath the thinner southern crust.

In order to assess how primitive the melt inclusions are and whether differences in S, Cl and Cu and other ore minerals are controlled by mantle or crustal processes, we need to see if these variations exist in the most primitive melt inclusions, e.g. $Fo > 80$. Thus, it is essential that instead of plotting Cu, S and Cl against K, which is susceptible to source composition and degree of melting, the authors plot these elements against Fo content of the phenocryst, using different symbols for the different phenocryst phases studied, although most are ol-hosted MIs. As shown by Wehrmann et al., the variations exist in olivine-hosted MIs fractionation-corrected to 91. Therefore, it looks like the controlling processes on these elements are mantle rather than crustal processes, for example degrees of melting and source composition.

Overall there is far too much information in the online appendices and the constant references to these appendices at critical parts of the manuscript make it very difficult to follow the arguments in the manuscript. In order to properly follow the manuscript, essential information needs to be moved from online files to the manuscript.

Manuscript

L 87-88. Judging from Supplementary Table 1 these criteria varied between the samples. For some samples the criteria of primitive was $Fo > 85$, for other $Fo > 74$. Because the goal of this screening is to select comparably primitive inclusions, this criteria should not vary, but should be fixed. It should also be the same for LA-ICP-MS and EMPA data. Usually inclusions in $Fo > 80$ are considered as relatively primitive. In this case, outliers in many plots - Llaima and Planchon inclusions in Fe-rich olivine ($Fo < 80$) and with very high Cu and low S - will be screened out. In any case, inclusions in olivine $Fo < 80$ cannot be considered as "the closest available approximation of mantle-derived magma composition". The authors claim in their rebuttal that such low Fo can result from re-equilibration of initially high Fo olivine. They however provide no evidence for their interpretation or against apparent and straightforward interpretation of these olivines as crystallized from evolved low Mg# magmas.

L 129-131. The outlying behaviour of Planchon and also Llaima is easily explained by low Mg# of olivine and evolved melt composition, not by enigmatic "particular local tectonic environment

leading to more rapid magma transfer through the crust”.

L 146-148. Planchon (yellow field) in the north also shows negative correlation K₂O and S, Cl.

L 166. I do not see trends of Llaima and Antuco on figure 3c.

L225-228. This is unclear and probably means that the assimilated intrusive rocks contain sulfide phase to explain strong positive K₂O-Cu correlation for, for example, Villarrica.

L 252-254. The authors probably meant K/Cl ratio, not K₂O concentrations. Supercritical fluids can carry significant amount of K and Nb and of course can exert a major control on their concentration in parental magmas.

L 272-273. Major conclusions of this study depend on this assumption.

L 295. This conclusion is highly model dependent, and the field of oceanic basalts is not shown. For example, Li and Audétat (2012) predict Cu/Ag as high as 10,000 already at 8% melting in presence of MSS. In my view Cu/Ag in SVZ is similar to mantle values and thus can be generated by any degree of melting in presence of liquid sulfide. The data deviating to Cu/Ag < 2000 can reflect MSS in the source or its early fractionation.

L353-354. The conclusion about H₂O is based on which observation(s)?

Comments to the authors' response to previous reviewer comments:

Lines 97-98: Cu and Pt have very different partition coefficients between silicate melt and sulfide. Therefore, they are not expected to show similar latitudinal or fractionation trends. See, for example, Pt and Cu systematics in MORB.

Line 115-116 and Fig. R1: Adding data from Wehrmann et al. is valuable contribution and shows consistency of these two data sets and acknowledges previous study.

Fig. R2, line 157 and following text:

It seems I have a fundamental disagreement with the authors on how to interpret variations of Mg# and incompatible elements in magmas.

The authors refer to Fig. R2 and claim that “overall negative correlation between both of these elements and host-Mg#” (line 160), “... fluid-mobile and -immobile elements are overall not correlated with host Mg#, except at Villarrica (and perhaps Antuco).” (line 185-186) and “...at Villarrica, Cl and S are positively correlated with Fo ($p < 0.05$; Fig. R2).” (line 252).

I have completely different view on Fig. R2. I see strong correlations between K₂O, Nb and Fo for most volcanoes with significant range of Fo, which suggest crystal fractionation control on all three parameters and no significant effect of Fe/Mg re-equilibration of host mineral composition or crustal assimilation on K₂O and Nb. Different volcanoes have different K₂O and Nb at the same Fo, which points to different initial K₂O and Nb in parental magmas. Thus, K₂O and Nb are dependent on both degree of fractionation AND on volcano (source). Olivine Fo is robust parameter of magma fractionation in this case. K₂O or Nb will also work as proxies of magma fractionation but only for genetically related melts from one volcano.

As the authors correctly note, Llaima and Planchon data do not show correlation with Fo. These olivines are the most evolved. Thus, they could trap the most evolved melts variably affected by crustal assimilation. Exceptionally high Cu in these two samples agrees with their evolved composition and crystallization at sulfide undersaturated conditions. Some scatter of Nb and K₂O does not necessarily require re-equilibration of originally more magnesian olivine.

Correlation of Cl and S with olivine Fo is not evident from Fig. R2. Listing correlation coefficients, critical correlation coefficients at $p = 0.05$ and showing lines of linear regression would help reader

to decide if these are strong correlations or weak.

Lines 173-178: "It is worthy to note though that K₂O concentrations in the melt inclusions seem to converge toward a similar value with increasing host olivine Mg# as shown on Supplementary Fig. 5, and both fluid-mobile (K₂O) and -immobile (Nb) elements exhibit the same distribution as a function of Mg#. This indicates that the variation in incompatible element concentrations is primarily induced by the incorporation of a crustal melt rather than a slab-derived fluid."

This is a completely confusing statement. These correlations reflect nothing but crystal fractionation: increasing K₂O and Nb with decreasing Fo.

Lines 226-228: The range of statistically significant correlation coefficients is quite large for large populations. The authors should provide correlation coefficients, which allow assessment of the correlations as "weak" ($r < 0.7$) or "strong" ($r > 0.7$).

Lines 348-353: Other labs detect this interference at typical ThO/Th=0.1-0.4%. It is worth perhaps to study this effect and use some true olivine reference materials.

**NCOMMS-20-19898-T** (by Grondahl and Zajacz)

**RESPONSES TO REVIEWER COMMENTS (ROUND 3)**

Please see our responses in blue following each comment.

Reviewer #3 (Remarks to the Author):

Comments on “Sulfur and chlorine budgets control the ore fertility of arc magmas” by Grondahl and
Zajacz

The manuscript represents an improvement over the previous version; nevertheless, many issues have
not been adequately addressed. The degassing of Cl-S in the south is unclear and not necessary to
generate low Cl-S magmas.

R1: We are showing trends of S and Cl decreasing with increasing degree of magma differentiation and
we discuss all possible reasons that could cause this (e.g., lines 222-309).

The discussion is extremely speculative. Also the criteria for "primitive" magma in this work is totally
arbitrary, simply a mess, in this manuscript.

R2: This is a completely new remark from this reviewer after two rounds of reviews. We and the other
two reviewers of the manuscript do not find the discussion speculative. Also we refer to the “most
primitive magmas” emplaced in upper crustal reservoirs in case of each volcano. We acknowledge in the
manuscript that some of them are slightly more evolved, and indeed, this observation is used in the
interpretation of the data (e.g., lines 203-206). However, in the perspective of ore fertility, what really
matters is the composition of the magma batches that reach the upper crust and this is what the studied
melt inclusions represent. Having said all the above, the fact that most (94 %) analyzed inclusions are in
olivine demonstrates that only minor to moderate degrees of crystal fractionation could have occurred
before the entrapment of the melt inclusions.

The major conclusion of the manuscript is that “the northern SVZ volcanoes generate world class
porphyry Cu deposits because the thick crust facilitates the generation of prolonged volatile-rich
magmatism where crucial volatile elements such as S and Cl are not significantly depleted in the magma
prior to emplacement at upper crustal depths.” It remains a contradiction in logic to me, why magmas
would not lose S and Cl when traversing thicker crust in the north than in the south. The authors’
provide two explanations for the behaviour of volatiles and ore minerals. They argue that increased
degassing takes place at lower to mid crustal depths beneath the southern SVZ volcanoes, based on the
“fluid/silicate melt partition coefficient of Cl ...strongly increasing with increasing pressure”. Why would
volcanoes in the south degas more than volcanoes in the north, particularly if the crust is thicker in the
north.

R3: We argue in the manuscript (lines 233-236) that the magma plumbing systems below the Southern
volcanoes are fed by initially more water-rich magmas due to the subduction of a major fracture zone on
the oceanic plate which likely facilitated much higher than average grade of serpentinization of the
subducted slab leading to increased water-influx into the mantle wedge underneath these volcanoes. The
observed trends are most pronounced at Villarrica, which lies directly above the subducting Valdivia
fracture zone.

The second reason given in the manuscript for the decrease of S and Cl is the assimilation of plutonic
roots. It is argued that the plutonic rocks have degassed their S and Cl, but not Cu and thus assimilation
of the plutonic roots in the south will cause S and Cl to go down and Cu to increase. It is not clear to me,
why older plutonic rocks will be preferably assimilated beneath the thinner southern crust than thicker
northern crust. It seems more logical that these processes are more likely (or at least as likely) to occur
under the thicker northern crust than beneath the thinner southern crust.

R4: We agree that the extent of assimilation is higher in the north (e.g., line 281, Supplementary Fig. 3 and
4). However, we argue that the deep plutonic roots in the south contain much less sulfide than those in
the north (consistently with all our observations; lines 337-339, 218-219, 231-232, Fig. 6). The result of
this is that the bulk silicate melt/lower crustal rock partition coefficients of chalcophile elements will be
significantly higher in the south as there are no residual sulfides withholding these. Accordingly, the
assimilated low-degree crustal melts will be Cu-rich in the south but Cu-poor in the north.

In order to assess how primitive the melt inclusions are and whether differences in S, Cl and Cu and
other ore minerals are controlled by mantle or crustal processes, we need to see if these variations exist
in the most primitive melt inclusions, e.g. $Fo > 80$. Thus, it is essential that instead of plotting Cu, S and Cl
against K, which is susceptible to source composition and degree of melting, the authors plot these
elements against Fo content of the phenocryst, using different symbols for the different phenocryst
phases studied, although most are ol-hosted MIs. As shown by Wehrmann et al., the variations exist in
olivine-hosted MIs fractionation-corrected to 91. Therefore, it looks like the controlling processes on
these elements are mantle rather than crustal processes, for example degrees of melting and source
composition.

R5: We have addressed the same comment in depth in our previous rounds of revisions. We reiterate our
disagreement with the idea that the host mineral Mg# is a better measure of the degree of differentiation
than the K_2O concentration in the melt within a single magmatic system. The reason is that the Mg# of
olivine can be very rapidly re-equilibrated by Fe-Mg interdiffusion whereas the K_2O concentration in the
melt inclusions is conservative. Furthermore, K_2O is more sensitive to crustal assimilation which plays a
very important role in magma differentiation in the Southern Volcanic Zone (e.g., lines 252-276,
Supplementary Figs. 3 and 4). The Mg# of the magmas in long-lived differentiating lower crustal reservoirs
could be buffered by a cumulate pile of mafic minerals and/or previously formed mafic country rocks
(lines 91-99, Supplementary Fig. 4, and responses during previous rounds of revision). Of course, source
composition may introduce variation in the K_2O content of primitive magmas as we acknowledge in
manuscript lines 290-296. For this reason, we DO NOT use K_2O for comparative analysis of the degree of
differentiation *between* various volcanic centers, only *within* individual ones.

We would like to note that the Wehrmann et al. 2014 article the reviewer refers to in the above comment
uses both K_2O and Mg# as differentiation indicator, and the observed trends in this paper are also more
pronounced when K_2O is used, as acknowledged by the authors.

Overall there is far too much information in the online appendices and the constant references to these
appendices at critical parts of the manuscript make it very difficult to follow the arguments in the
manuscript. In order to properly follow the manuscript, essential information needs to be moved from
online files to the manuscript.

R6: We had such references only in a few places in the manuscript. Nevertheless, to address this request,
we have moved most of the Supplementary Discussion into the main text in this round of revisions (e.g.,
lines 256-273 and Figs. 3 and 4 captions).

**Manuscript**

L 87-88. Judging from Supplementary Table 1 these criteria varied between the samples. For some
samples the criteria of primitive was $Fo > 85$, for other $Fo > 74$. Because the goal of this screening is to
select comparably primitive inclusions, this criteria should not vary, but should be fixed. It should also be
the same for LA-ICP-MS and EMPA data. Usually inclusions in $Fo > 80$ are considered as relatively
primitive. In this case, outliers in many plots - Llama and Planchon inclusions in Fe-rich olivine ($Fo < 80$)
and with very high Cu and low S - will be screened out. In any case, inclusions in olivine $Fo < 80$ cannot be
considered as “the closest available approximation of mantle-derived magma composition”. The authors
claim in their rebuttal that such low Fo can result from re-equilibration of initially high Fo olivine. They
however provide no evidence for their interpretation or against apparent and straightforward
interpretation of these olivines as crystallized from evolved low Mg# magmas.

R7: Nowhere do we state that the goal of the pSMI screening is to ‘select comparably primitive inclusions’.
Instead, we consider such a screening to have already been employed simply by the presentation of our
full dataset containing SMI with $\leq \sim 60$ wt% SiO_2 , the vast majority (94 %) of which are olivine-hosted. As
we explain (lines 106-108), we applied the pSMI filter to further select the most primitive SMI available *at*
*each location* — this *should* vary location to location because no two systems are expected to have
identical magmatic histories. The “closest available approximation of initial mantle-derived magma
compositions” is, by its very definition, whatever the least-evolved SMI or sample is, regardless of its
relation to a given Mg# threshold (e.g., $Fo = 80$).

The claim that that we provide no evidence that measured Fo systematics represent more than pure
crystal fractionation is simply untrue. We provide specific reasoning related to our samples in lines 173-
189 of the responses to previous round of comments.

As the reviewer notes, we do explain how Fo can be modified by re-equilibration with the surrounding
melt, but for the reviewer to focus solely on this point ignores the preceding lines (179-213) in our
responses to the previous round of comments) where we explain at length how Mg#'s and other major
elements can be affected both in crystallizing phases and their host magmas by open system processes
which are very difficult to precisely quantify and disentangle, and will impact melt composition *prior* to

arrival in the upper crust and subsequent crystallization. Please note that we have attempted to clarify
this further in lines 91-99 and Supplementary Fig. 4)

L 129-131. The outlying behaviour of Planchon and also Llaima is easily explained by low Mg# of olivine
and evolved melt composition, not by enigmatic “particular local tectonic environment leading to more
rapid magma transfer through the crust”.

R8: We do not understand what the reviewer means by ‘evolved melt composition.’ SMI from both
locations have fairly high FeO ($\geq \sim 9$ wt%), MgO ($\geq \sim 4$ wt%), and CaO ($\geq \sim 8-9$ wt%), and low SiO₂ ($\leq \sim 55$ wt
128 %), and certainly do not appear to be more evolved than other locations. Regarding olivine Mg#, we have
129 explained multiple times that this is susceptible to deviation from simple closed-system crystallization-
130 dominated controls (e.g., responses R5 and R7 above).

L 146-148. Planchon (yellow field) in the north also shows negative correlation K₂O and S, Cl.

R9: Indeed, and this is discussed in the manuscript (lines 150-154, 193-198, 282-284).

L 166. I do not see trends of Llaima and Antuco on figure 3c.

We agree that there is a lack of clear trends here, and have rephrased the existing text to reflect this
wording specifically (now lines 191-193).

L225-228. This is unclear and probably means that the assimilated intrusive rocks contain sulfide phase
to explain strong positive K₂O-Cu correlation for, for example, Villarrica.

R10: It is unlikely that any non-negligible sulfide phase would be consumed during the melting of the lower
crustal rock because sulfide solubility in felsic melts is very low. Therefore, sulfide would remain a residual
phase and in fact decrease the bulk silicate melt/lower crustal rock partition coefficient of Cu — the
opposite effect relative to what the reviewer refers to in this comment.

L 252-254. The authors probably meant K/Cl ratio, not K₂O concentrations. Supercritical fluids can carry
significant amount of K and Nb and of course can exert a major control on their concentration in
parental magmas.

R11: Thank you – we have amended these lines (now lines 303-304) to clarify that we are referring to
“evolving K₂O concentrations and K₂O/Cl ratios during differentiation”. Furthermore, the SVZ magmas
exhibit pronounced negative Nb-Ta anomalies, which demonstrates that the relative mobility of K is much
higher than Nb in the slab derived fluids in this setting.

L 272-273. Major conclusions of this study depend on this assumption.

R12: Please see our explanation above (response R3) and lines 222-251 concerning why we think that
more pronounced deep degassing in the south is consistent with the known tectono-magmatic framework
of the subduction zone and experimental constraints on volatile solubilities in magmas.

L 295. This conclusion is highly model dependent, and the field of oceanic basalts is not shown. For
example, Li and Audetat (2012) predict Cu/Ag as high as 10,000 already at 8% melting in presence of
MSS. In my view Cu/Ag in SVZ is similar to mantle values and thus can be generated by any degree of
melting in presence of liquid sulfide. The data deviating to Cu/Ag<2000 can reflect MSS in the source or
its early fractionation.

R13: We have now specified that by 'oceanic basalts', we mean MORB which we do show on Figure 6.

Regarding the modeling exercise of Li and Audetat (2012) please note also that Cu/Ag only reached 10,000
in the case where the sulfide fraction was entirely MSS, with no sulfide liquid present. For an idea of the
importance of liquid sulfide in their model, note that their 100% and 50% sulfide liquid (0 % and 50 %
MSS) curves are nearly identical to one another and have nearly constant Cu/Ag during melting, in strong
contrast to the MSS-only curve. This is similar to our model wherein varying degrees of melting (~6–12 %)
result only in a factor of less than ~2 variation in the Cu/Ag ratio of the mantle melt, even though we
invoke a smaller proportion of sulfide liquid (90 % MSS : 10 % sulfide liquid). This similarity suggests
instead that our model results are rather robust, and please also note that our model uses updated values
for sulfide-silicate partition coefficients reported by Li and Audetat (2015). We therefore stand by our
model as currently presented.

L353-354. The conclusion about H2O is based on which observation(s)?

R14: This conclusion is based on the importance of preserving degassing-sensitive S and Cl budgets, and
therefore avoiding early fluid saturation which would be more likely to occur in unusually water-rich
magmas.

**Comments to the authors' response to previous reviewer comments:**

Lines 97-98: Cu and Pt have very different partition coefficients between silicate melt and sulfide.
Therefore, they are not expected to show similar latitudinal or fractionation trends. See, for example, Pt
and Cu systematics in MORB.

R15: The difference in partition coefficients is that Pt is more strongly partitioned into sulfide than Cu. We
have now specified (line 123-124) that we would expect a pattern of depletion in a manner similar to Cu.
Although we did not feel that the original phrasing implied that the depletion of Cu and Pt would be
expected to be identical, we hope that any ambiguity here is now resolved.

Line 115-116 and Fig. R1: Adding data from Wehrmann et al. is valuable contribution and shows
consistency of these two data sets and acknowledges previous study.

R16: We have now included this data on the manuscript figure (Fig. 3).

Fig. R2, line 157 and following text:

It seems I have a fundamental disagreement with the authors on how to interpret variations of Mg# and
incompatible elements in magmas.

R17: This disagreement appears to be rooted in the reviewer's steadfast adherence to closed-system
crystal fractionation models, which we consider an over-simplification that should be treated with caution
and can be avoided by the use of incompatible element proxies for differentiation. We have addressed
this at length in previous rounds of responses, responses R5, R7, R18, and R19 here, manuscript lines 91-
99, and Supplementary Fig. 4.

The authors refer to Fig. R2 and claim that "overall negative correlation between both of these elements
and host-Mg#" (line 160), "... fluid-mobile and -immobile elements are overall not correlated with host
207 Mg#, except at Villarrica (and perhaps Antuco)." (line 185-186) and "...at Villarrica, Cl and S are positively
correlated with Fo ($p < 0.05$; Fig. R2)." (line 252).

I have completely different view on Fig. R2. I see strong correlations between K₂O, Nb and Fo for most
volcanoes with significant range of Fo, which suggest crystal fractionation control on all three
parameters and no significant effect of Fe/Mg re-equilibration of host mineral composition or crustal
assimilation on K₂O and Nb. Different volcanoes have different K₂O and Nb at the same Fo, which points
to different initial K₂O and Nb in parental magmas. Thus, K₂O and Nb are dependent on both degree of
fractionation AND on volcano (source). Olivine Fo is robust parameter of magma fractionation in this
case. K₂O or Nb will also work as proxies of magma fractionation but only for genetically related melts
from one volcano.

R18: We see a clear correlation only in the case of Villarrica and perhaps Maipo, not most volcanoes. It is
true that different volcanoes have different K₂O and Nb at the same host Mg# and we have never claimed
otherwise. This could be the result of the following two broad processes operating in sequence, each
acknowledged and addressed in previous rounds of revision:

1. Different degree of melting of a mantle source (this forms an integral part of our interpretation; lines
290-292, Figure 6) with *inter*-volcano variability in slab fluid input to the mantle source. This would
primarily apply to K₂O rather than Nb in our opinion (we indeed suggest this to be the reason for the inter-
volcano variability of S and Cl concentrations when the most primitive SMI available are compared from
each; lines 290-296). Please see also response R11 above.

2. Varying degree of deep crustal assimilation while the Mg# of the magma is buffered by a cumulate pile
or mafic country rock. In our opinion, this is the best and perhaps only feasible explanation for the 2-3
229 times increase in highly incompatible element concentrations in the melt (e.g., Fig. R2 from the previous
revision round; K₂O in manuscript Fig. 4; lines 256-261; Supplementary Figs. 3 and 4) while the magma
remains within the olivine stability field and the olivine often exhibits a narrow range of Mg#. This degree
of increase in incompatible element concentrations would require 50 – 70% crystallization of the magma
if it were solely to be explained by crystal fractionation as the reviewer suggests. Over such a wide range
of crystallization, the major element composition of the magma, the stable phase assemblage (e.g.,
olivine-bearing or not), and the Mg# of any surviving non-re-equilibrated ferromagnesian minerals should

show a much broader range than observed in this study (and many others in arc settings worldwide). This
process and the potential diffusive Fe-Mg equilibration of the host mineral after melt inclusion
entrapment explain why there is no clear intra-volcano correlation between host Mg# and incompatible
element concentrations in the studied volcanoes other than Villarrica and perhaps Maipo.

As the authors correctly note, Llaima and Planchon data do not show correlation with Fo. These olivines
are the most evolved. Thus, they could trap the most evolved melts variably affected by crustal
assimilation. Exceptionally high Cu in these two samples agrees with their evolved composition and
crystallization at sulfide undersaturated conditions. Some scatter of Nb and K₂O does not necessarily
require re-equilibration of originally more magnesian olivine.

R19: The supposition that olivine from these volcanoes have trapped especially ‘evolved compositions’
can easily be checked, and by inspection it is clear that the SMI are no more evolved in their major element
composition than the general dataset from the other volcanoes. This again begs the question as to
whether major and trace elements have been decoupled from a simple crystallization-driven closed
system relationship in the melts from which olivine is crystallizing. As we explained in lines 190-213 of our
responses to the previous round of comments, such a *decoupling via assimilation of or re-equilibration*
*with, for example, volumetrically-dominant solids within crystal mushes (see the several references*
*provided previously and additionally Klaver et al. 2018, EPSL and Lissenberg & MacLeod 2016, JPet) can*
*modify the melt composition prior to upper crustal olivine crystallization and this is a distinct process not*
*recorded by the later formed olivine phenocrysts (i.e., the SMI hosts). Again, this is why we are hesitant to*
*directly interpret low-Fo olivine as simply the product of relatively extensive closed-system crystal*
*fractionation, and why we prefer incompatible trace element proxies which are sensitive to the full variety*
*of processes operating during differentiation — and, when decoupled from Cl and S, allow the inference*
*of non-crystallization processes. The use of the more general K₂O to show how differentiation within each*
*individual volcano affects other elements like Cu, S and Cl is therefore better suited for summary diagrams*
*as used in the manuscript. This consideration is discussed at lines 91-99 and Supplementary Fig. 4; please*
*see also response R17 here, and lengthy discussions during previous rounds of responses.*

Correlation of Cl and S with olivine Fo is not evident from Fig. R2. Listing correlation coefficients, critical
correlation coefficients at p=0.05 and showing lines of linear regression would help reader to decide if
these are strong correlations or weak.

R20: Please see response R23 below.

Lines 173-178: “It is worthy to note though that K₂O concentrations in the melt inclusions seem to
converge toward a similar value with increasing host olivine Mg# as shown on Supplementary Fig. 5, and
both fluid-mobile (K₂O) and -immobile (Nb) elements exhibit the same distribution as a function of Mg#.
This indicates that the variation in incompatible element concentrations is primarily induced by the
incorporation of a crustal melt rather than a slab-derived fluid.”

This is a completely confusing statement. These correlations reflect nothing but crystal fractionation:
increasing K₂O and Nb with decreasing Fo.

R21: Please note that the cited sentence is from our last response letter and not the manuscript. We think
our previous responses at R5, R17, R18 and R19 adequately address this comment.

Lines 226-228: The range of statistically significant correlation coefficients is quite large for large
populations. The authors should provide correlation coefficients, which allow assessment of the
correlations as “weak” ($r < 0.7$) or “strong” ($r > 0.7$).

R23: Although we maintain that our use of the term correlation is in keeping with common usage in our
field, we have replaced this word with alternate phrasing that has less restrictive statistical definitions
(lines 170-173, 323-326).

Lines 348-353: Other labs detect this interference at typical $\text{ThO}/\text{Th} = 0.1\text{-}0.4\%$. It is worth perhaps to
study this effect and use some true olivine reference materials.

R24: We can only truly speak for the performance of our instrument, and in the referenced lines we
demonstrated by specific testing that Sc concentrations in olivine were not sensitive to even very high
nominal oxide production rates ($\text{ThO}/\text{Th} \sim 1\%$). We do agree that one must always be careful of these
potential interferences, but we maintain that this issue did not compromise our measurements. Also, we
would like to reiterate that Sc is not utilized at all in our interpretations.

REVIEWER COMMENTS

Reviewer #1 (Remarks to the Author):

Dear Editor,

I went through all the reviewer comments and corresponding replies, and I carefully read the current version of the manuscript. The result are rather mixed feelings.

With regard to the intense discussion about whether Mg# or K₂O is a better indicator of magma differentiation I understand the arguments of both parties. The reviewer does not believe in K₂O as an indicator of magma differentiation because it may reflect other processes, such as the degree of partial melting. This is certainly true. However, the authors stated clearly that they use K₂O as differentiation indicator only within individual volcanic systems, so I think they sufficiently dealt with that criticism. Furthermore, I agree with the authors that two- to three-fold increases in incompatible elements within relatively small ranges of olivine Mg numbers cannot be explained by closed-system fractionation, as olivine does not precipitate over such large crystallization intervals. Therefore, with regard to this issue I am more on the author's side.

However, I feel that the manuscript is now terribly bloated due to the several rounds of revisions it has gone through, and I agree with the reviewer that large parts of the discussion are very speculative. There are so many lengthy discussions about potential processes that may have occurred at depth that the main message is almost lost. In my opinion, the story would be already complete if the authors were able to demonstrate that the metal and volatile contents of the investigated melt inclusions have not been affected by upper crustal magma chamber processes. However, with regard to this issue the manuscript is very unclear: at some locations the authors argue that all the investigated melt inclusions were trapped in the upper crustal magma chambers (e.g., lines 85, 96-97, 135-138, 194), whereas at other locations they argue that the melt inclusions record processes that occurred in the middle or lower crust (e.g., line 86, 88-89, 103, 174-175, 189). This does not fit together, and it even seems that the authors have no clue at which depth their inclusions formed. The argument that melt inclusions that form in upper crustal magma chambers are still able to record the evolution of volatiles at greater depth (lines 85-89) is implausible, as magmas likely lose volatiles during their ascent (on lines 225-254 the authors argue that the magmas were fluid-saturated already at depth). Therefore, in my view the most important task is to demonstrate that the measured volatile contents were not affected by degassing in the upper crustal magma chambers. All the rest (now ca. 75% of the manuscript) is of secondary importance.

NCOMMS-20-19898-T (by Grondahl and Zajacz)

Responses to reviewer comments (R4)

Please see our responses in blue italic inserted after the reviewer comments.

Reviewer #1 (Remarks to the Author):

I went through all the reviewer comments and corresponding replies, and I carefully read the current version of the manuscript. The result are rather mixed feelings.

With regard to the intense discussion about whether Mg# or K₂O is a better indicator of magma differentiation I understand the arguments of both parties. The reviewer does not believe in K₂O as an indicator of magma differentiation because it may reflect other processes, such as the degree of partial melting. This is certainly true. However, the authors stated clearly that they use K₂O as differentiation indicator only within individual volcanic systems, so I think they sufficiently dealt with that criticism. Furthermore, I agree with the authors that two- to three-fold increases in incompatible elements within relatively small ranges of olivine Mg numbers cannot be explained by closed-system fractionation, as olivine does not precipitate over such large crystallization intervals. Therefore, with regard to this issue I am more on the author's side.

We are glad the reviewer agrees with us on this point. To address the following comment, we have removed the extensive discussion of this issue to streamline the manuscript and added a more succinct paragraph instead at the beginning of the discussion (line 150-157) to explain our choice of the use of K₂O as an indicator of the degree of magma differentiation.

However, I feel that the manuscript is now terribly bloated due to the several rounds of revisions it has gone through, and I agree with the reviewer that large parts of the discussion are very speculative. There are so many lengthy discussions about potential processes that may have occurred at depth that the main message is almost lost. In my opinion, the story would be already complete if the authors were able to demonstrate that the metal and volatile contents of the investigated melt inclusions have not been affected by upper crustal magma chamber processes.

We removed several sections which could be seen as speculative or disruptive for the flow of the text. This includes the discussion of the choice of differentiation indicator at the end of the first paragraph after Fig. 1, the elaboration on the likelihood of the intra-volcano variation of K/Cl ratios introduced during magma generation (lines 295-301) and the extensive discussion in support of the broadly accepted significance of crustal assimilation (lines 251-268), which is now covered briefly at lines 150-157 and in more details in the caption of the Supplementary Figures 3 and 4.

However, with regard to this issue the manuscript is very unclear: at some locations the authors argue that all the investigated melt inclusions were trapped in the upper crustal magma chambers (e.g., lines 85, 96-97, 135-138, 194), whereas at other locations they argue that the melt inclusions record processes that occurred in the middle or lower crust (e.g., line 86, 88-89, 103, 174-175, 189). This does not fit together, and it even seems that the authors have no clue at which depth their inclusions formed. The argument that melt inclusions that form in upper crustal magma chambers are still able to record the evolution of volatiles at greater depth (lines 85-89) is implausible, as magmas likely lose volatiles

during their ascent (on lines 225-254 the authors argue that the magmas were fluid-saturated already at depth). Therefore, in my view the most important task is to demonstrate that the measured volatile contents were not affected by degassing in the upper crustal magma chambers. All the rest (now ca. 75% of the manuscript) is of secondary importance.

We agree with the reviewer that some of the listed statements could be seen as self-contradictory. We are fully convinced that the variations in volatile element concentrations developed at greater depths otherwise the observed correlations with K_2O would have to be coincidental, which seems rather unlikely (i.e. variable extent of shallow degassing would yield random scatter on K_2O vs. volatile element diagrams keeping in mind that the intra-volcano variation of K_2O itself must have developed at greater depths because of thermal constraints, see e.g. lines 150-157). To clarify this point we also included a triple arrow on Fig. 3b showing the vectors induced by the variety of processes in consideration. An additional strong argument against shallow degassing is that significant loss of Cl from mafic melts by degassing at upper crustal pressures is not possible as discussed in the manuscript (lines 217-223, 847-852). An additional discussion of this issue is included in the supplementary material (lines 36-54).

Within the emerging view of trans crustal magmatic systems, most of the compositional diversity of continental arc magmas develops via magma differentiation at lower to mid crustal depths, and derivative melts are transported to upper crustal magma reservoirs often without significant crystal cargo. Olivine is not suitable for geobarometry and also facilitates rapid post-entrapment diffusive loss of H_2O , thus we cannot provide an independent constraint on the entrapment depths of the melt inclusions. Nevertheless, we think that our key supporting arguments above along with the fact that olivine is the first crystallizing phase in the studied magmas leave little doubt that the negative correlations between K_2O and the volatile Cl and S must have developed at mid to lower crustal depths. Additional degassing upon magma ascent from deep crustal levels would not affect Cl concentration significantly due to the decreasing fluid/melt partition coefficients of Cl with decreasing pressure, and would only serve to erase any systematic trends on the K_2O vs S diagram because shallow S degassing would affect these magmas largely independent of their K_2O content.

To avoid any confusion with regards to this issue, we have removed or rephrased all statements flagged by the reviewer as contradictory, most of which were introduced in response to previous review comments (i.e. lines 85, 96-97, 135-138, 194 in Revision 3 as listed by the reviewer, seen at lines 66, 77-78, 116-117, 174 in the newly revised version).